# Patterns and ecological drivers of viral communities in acid mine drainage sediments across Southern China

Shaoming Gao [1], David Paez-Espino[2], Jintian Li[3], Hongxia Ai[1], Jieliang Liang[3], Zhenhao Luo[1], Jin Zheng[3], Hao Chen[4], Wensheng Shu [3] & Linan Huang [1]✉

Recent advances in environmental genomics have provided unprecedented opportunities for the investigation of viruses in natural settings. Yet, our knowledge of viral biogeographic patterns and the corresponding drivers is still limited. Here, we perform metagenomic deep sequencing on 90 acid mine drainage (AMD) sediments sampled across Southern China and examine the biogeography of viruses in this extreme environment. The results demonstrate that prokaryotic communities dictate viral taxonomic and functional diversity, abundance and structure, whereas other factors especially latitude and mean annual temperature also impact viral populations and functions. In silico predictions highlight lineage-specific virus-host abundance ratios and richness-dependent virus-host interaction structure. Further functional analyses reveal important roles of environmental conditions and horizontal gene transfers in shaping viral auxiliary metabolic genes potentially involved in phosphorus assimilation. Our findings underscore the importance of both abiotic and biotic factors in predicting the taxonomic and functional biogeographic dynamics of viruses in the AMD sediments.

[1] School of Life Sciences, Sun Yat-sen University, Guangzhou 510275, PR China. [2] DOE Joint Genome Institute, Lawrence Berkeley National Laboratory, Berkeley, CA 94720, USA. [3] School of Life Sciences, South China Normal University, Guangzhou 510631, China. [4] School of Environment and Energy, South China University of Technology, Guangzhou 510006, China. ✉email: eseshln@mail.sysu.edu.cn

Microorganisms are the most phylogenetically diverse and widespread form of life on Earth[1]. Unraveling the processes that generate and underlie microbial biodiversity across space and time is critical for predicting the dynamics of microbial communities in the environment[2,3]. Gene surveys, especially those utilizing high throughput sequencing technologies, have advanced our understanding of the biogeographic patterns of microbes in nature, revealing significant roles of contemporary environmental variation or historical contingency in shaping their large-scale ecological ranges[4]. More recently, advances in metagenomic sequencing technologies and bioinformatics have moved microbial biogeography forward, allowing the examination of functional trait variation in their natural settings and the evolutionary and ecological processes creating and maintaining the biogeographic patterns[5,6]. Collectively, these efforts have greatly furthered our understanding of the mechanisms shaping microbial biodiversity on the planet.

Viruses are key entities in natural microbial assemblies, impacting prokaryotic population size through lysis[7], reprogramming host metabolism with auxiliary metabolic genes (AMGs)[8], and shaping microbial evolution via horizontal gene transfers (HGTs)[9]. However, viral ecology studies have been hampered by an absence of universal marker genes and thus were traditionally dependent on cultivation-based methods[10]. More recently, meta-omics approaches have been applied to explore viral diversity in the environment[11], uncovering high viral diversity with little similarity to previously recognised viruses[12]. Despite these progresses, the biogeographic variation of viruses in ecosystems remains largely unstudied. The marine environments have been the focus of several studies of viral biogeography, revealing patterns whereby viral communities are passively transported on oceanic currents and locally structured by environmental conditions[13], and the existence of specific ecological zones throughout the global ocean, with epipelagic waters and the Arctic as hotspots for viral biodiversity[14]. Our current understanding of viral biogeography stems from these pioneering studies.

The reduced-complexity prokaryotic communities in extreme environments have served as models for the study of microbial community structure and function[15,16]. The relatively low species richness, broad range and steep gradients of geochemical variables promise more straight-forward establishment of ecological patterns and underlying mechanisms. The diversity and community dynamics of viruses in extreme environments such as the Atacama Desert[17], cryosphere[18,19], acid mine drainage (AMD) environment[20,21], and Earth's subsurface[7,22,23] have recently been investigated through meta-omics approaches; yet, extensive sampling and analysis of viral communities across large geographic scales to resolve their ecological distribution patterns and drivers have not been conducted. Here we strive to address this knowledge gap by utilizing a massive metagenomic data set generated from 90 AMD sediments sampled across Southern China (Fig. 1a). Extensive recovery of viral and prokaryotic genomes was performed and the results were analysed with a comprehensive set of metadata on geochemistry, geographic location and climate variables for each sample[24], to quantify the effects of both biotic (prokaryotic hosts) and abiotic factors on the viral assemblages in this extreme ecosystem.

## Results

**Viral diversity in the AMD sediments**. Metagenomic sequencing was conducted on the 90 sediment samples taken from geographically separated and geochemically diverse AMD environments[24]. Assemblies from the metagenomes were screened using a viral protein families-based pipeline[25], VirSorter v1.0.6[26] and

CheckV v0.6.0[27] and manually curated to predict 11,112 putative viral genomes that ranged between 10 - 350 kb with ~94% from 10 to 50 kb in size (Fig. 1b and Supplementary Data 1). We identified a total of 5,678 potential viral populations (viral operational taxonomic units, vOTUs), which are suggested to approximately represent species-level taxonomy[12], and 143,610 viral protein clusters (PCs) that help organise the dominant unknown sequence space[13] (Fig. 1c). The number of vOTUs and viral PCs in each sample ranged from 537 to 3,199 and 6,628 to 52,631, respectively (Supplementary Data 2). Despite such a broad range in viral taxonomic and functional richness across all samples, the cumulative curves of vOTUs and PCs were saturated, indicating that viral communities in the AMD sediments were relatively adequately sampled (Fig. 1c).

Taxonomic analyses of the 5,678 viral population genomes against the NCBI Viral RefSeq v201 database showed that the vast majority (96.0%) of vOTUs could not be assigned taxonomy through reticulate classification (vConTACT2)[28], while 66.1% of vOTUs could be annotated at the family level using the LCA algorithm[29] (Fig. 1c). Most classified viruses were resolved as one of the three families (*Myoviridae*, *Siphoviridae*, and *Podoviridae*) in the *Caudovirales* order (Fig. 1d and Supplementary Data 3). Comparisons of the predicted viral proteins against the eggNOG database[30] and VOG database revealed that most viral proteins from the AMD sediments were uncharacterised, with the annotated proteins enriched in information storage and processing (COG categories ABJKL) and virus replication (VOG category Xr) or virus function beneficial for the host (VOG category Xh) (Fig. 1e).

**Distribution patterns of viral diversity and functions**. To explore the variability in viral populations and functions across the AMD sediments, pairwise Pearson's correlations were used to uncover relationships between viral communities and other biotic and abiotic factors. Prokaryotic community structure in the sediments was resolved by extensive reconstruction and dereplication of bacterial and archaeal genomes from metagenomes, and the results were highly similar to those from the 16 S rRNA gene amplicon analysis[24] (Supplementary Fig. 1). The prokaryotic richness, estimated as the number of prokaryotic metagenome-assembled genomes (MAGs) in each sample (Supplementary Data 1), was found to be most relevant to the number of viral populations (Pearson's $r = 0.89$, $P < 0.001$) and functions (Pearson's $r = 0.82$, $P < 0.001$) (Fig. 2a). Meanwhile, overall viral taxonomic and functional richness increased toward the equator and were both negatively correlated with electronic conductivity (EC). Significant positive correlations were observed between viral abundance and ferric iron (Pearson's $r = 0.30$, $P < 0.05$), as well as between viral functional abundance and Fe (Pearson's $r = 0.29$, $P < 0.05$). We further evaluated the dependence of viral taxonomic and functional distributions on different factors by correlating dissimilarities of viral taxonomic and functional community composition with those of abiotic variables. Results showed that mean annual temperature (MAT) and Fe were the strongest correlates of both viral taxonomic and functional dissimilarities, which also increased with increasing differences in mean annual precipitation (MAP), pH, ferric iron, sulphate, and distance from the equator of the AMD sediments (Fig. 2a). Furthermore, Mantel test analysis revealed significant correlations between prokaryotic dissimilarity and viral taxonomic (Mantel's $r = 0.96$, $P < 0.001$) and functional dissimilarities (Mantel's $r = 0.95$, $P < 0.001$) across all samples.

To examine whether geographic distance may influence viral distributions, principal coordinate analyses (PCoA) were used to assess the degree of segregation of the viral communities. We

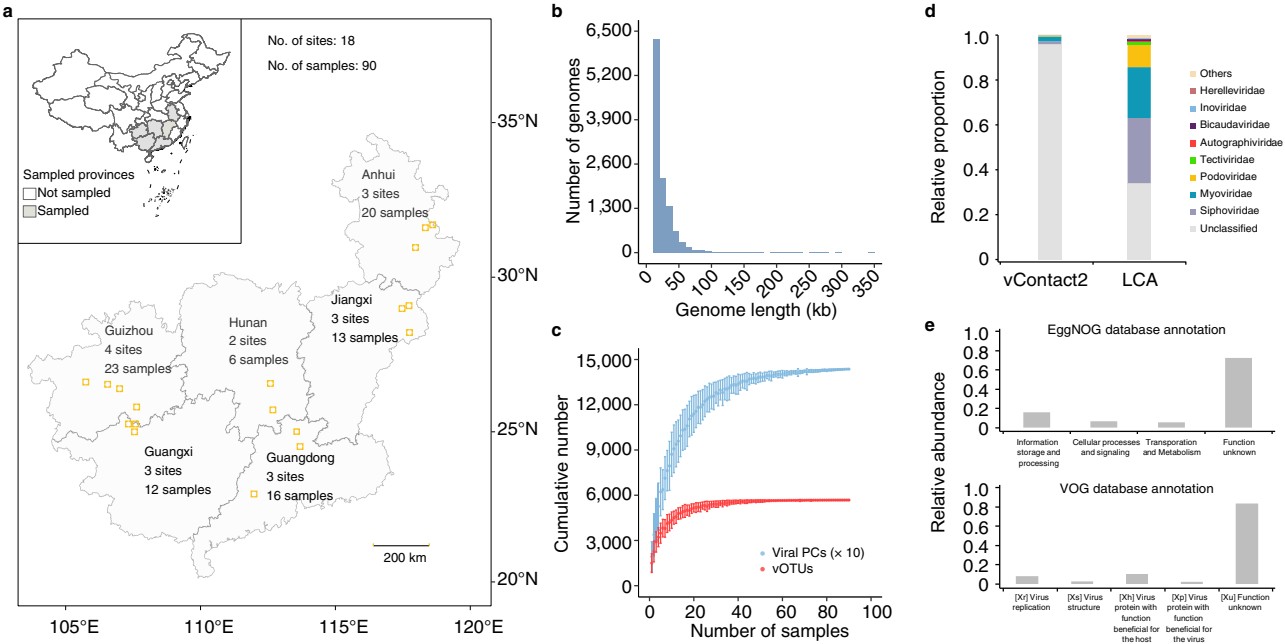

**Fig. 1 Overview of acid mine drainage (AMD) sediment viruses. a** Geographic distribution of collected AMD sediment samples. The provinces from which AMD sediments were sampled are presented in gray. All sampled AMD sites ($n = 18$) are marked by orange squares. **b** Histogram showing the distribution of viral genome size. **c** Accumulation curve of viral operational taxonomic units (vOTUs, red) and viral protein clusters (PCs, blue) in the AMD sediment metagenomes. Dots represent the average number of vOTUs and PCs for all combinations of a given number of samples, and error bars represent the range. The numbers of viral PCs were divided by ten for better visualization. **d** Bar graphs showing the relative proportion and taxonomy of vOTUs based on reticulate classification method (vContact2) and Lowest Common Ancestor (LCA) algorithm. **e** Relative abundances of viral functions in the AMD sediments as annotated by eggNOG v5.0.0 database and VOG database. All COG categories are grouped into four types, including information storage and processing (COG categories A, RNA processing and modification; B, chromatin structure and dynamics; J, translation, ribosomal structure and biogenesis; K, transcription; L, replication, recombination and repair), cellular processes and signaling (D, cell cycle control, cell division, chromosome partitioning; M, cell wall/membrane/envelope biogenesis; N, cell motility; O, posttranslational modification, protein turnover, chaperones; T, signal transduction mechanisms; U, intracellular trafficking, secretion and vesicular transport, V, defence mechanisms; W, extracellular structures; Y, nuclear structure; Z, cytoskeleton), metabolism and transportation (C, energy production and conversion; E, amino-acid transport and metabolism; F, nucleotide transport and metabolism; G, carbohydrate transport and metabolism; H, coenzyme transport and metabolism; I, lipid transport and metabolism; P, inorganic ion transport and metabolism; Q, secondary metabolites biosynthesis, transport and catabolism), and unknown functions (S, unknown; Unannotated proteins). Source data are provided in the Source Data file.

observed a separation of viral taxonomic and functional structure for the 90 AMD sediment samples, with a similar distribution within the same site (Fig. 2b, c). In support of this, significant negative distance-decay relationships (DDRs) were observed across all samples based on the Bray-Curtis similarities (1 - dissimilarity) of viral taxonomic (slope = −0.10, P < 0.001) and functional (slope = −0.09, P < 0.001) structure. Furthermore, the slopes of the DDRs depended on spatial scale. Specifically, the overall slope was significantly shallower than the slopes within a local scale (pairwise distance ≤ 1 km) but steeper than the slopes within a regional scale (pairwise distance > 1 km) (Fig. 2d, e).

**Ecological drivers of viral taxonomic and functional community structure.** Having illustrated the roles of individual factors in shaping viral taxonomic and functional diversity and distributions, we next sought to discern the causality and quantify the direct and indirect effects of the drivers using structural equation modeling (SEM). The final SEM models provided satisfactory fit to our data compared with the priori models (Supplementary Fig. 2), as suggested by the P-values (Chi-squared test) and root mean square error of approximation (RMSEA) (Fig. 3). Specifically, the hypothesised direct effects of pH on prokaryotic diversity and community structure in the priori models were not observed in our final SEM models. For viral communities, we did not find significant impacts of viral taxonomic and functional abundance on their composition, suggesting discrepancies

between our priori predictions and the final models (Fig. 3 and Supplementary Fig. 2). On the other hand, our final SEM models were consistent with the Pearson's correlation results. Distance from the equator probably had impacts on the number of vOTUs and viral PCs in different samples through its direct negative effect on MAP (r = −0.32, P < 0.01), or prokaryotic richness (r = −0.42, P < 0.001) which was the most influential variable directly related to viral taxonomic (r = 0.86, P < 0.001) and functional richness (r = 0.81, P < 0.001). The SEM models also revealed that pH and MAT had some direct effect on viral taxonomic and functional richness (Fig. 3a, b).

The prokaryotic richness had negative impacts on viral taxonomic (r = −0.33, P < 0.001) and functional (r = −0.16, P < 0.001) composition. Meanwhile, prokaryotic composition, which was positively and directly affected by MAT (r = 0.72, P < 0.001), distance from the equator (r = 0.58, P < 0.001) and prokaryotic abundance (r = 0.29, P < 0.001), was found to drive both viral taxonomic (r = 0.94, P < 0.001) and functional (r = 0.96, P < 0.001) composition. Unexpectedly, the abundances of viral populations and functions were negatively related to the abundance of prokaryotes, which was negatively driven by pH (r = −0.32, P < 0.001) and MAP (r = −0.25, P < 0.001), and positively associated with MAT (r = 0.93, P < 0.001) and distance from the equator (r = 0.74, P < 0.001). Additionally, both MAP and prokaryotic richness affected the abundances of viral populations and functions, with increased abundance associated

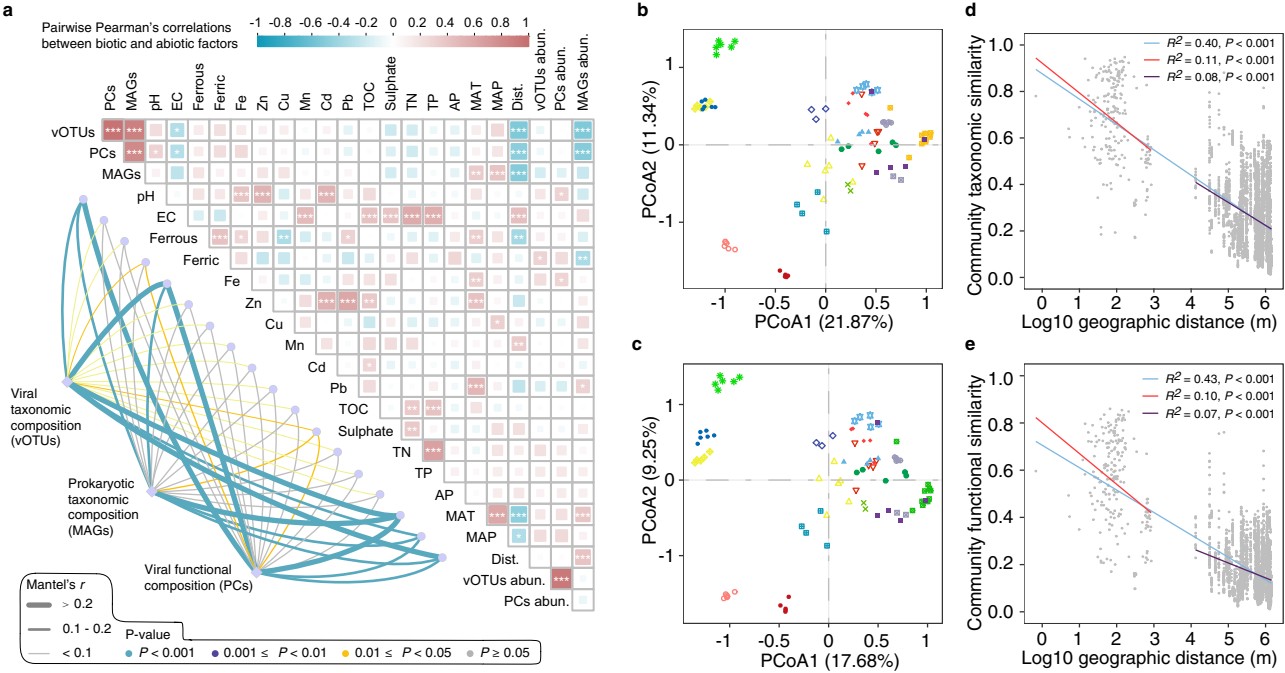

**Fig. 2 Dynamics of viral populations and functions. a** Pairwise comparisons of the biotic and abiotic variables. The color gradient in the heatmap denotes Pearson's correlation coefficients and the asterisk indicates two-tailed test of Pearson's statistical significance adjusted using the Benjamini and Hochberg false discovery rate controlling procedure. *$P < 0.05$, **$P < 0.01$ and ***$P < 0.001$. Edge width corresponds to the Mantel's r statistic for the corresponding distance correlations, and edge color denotes the statistical significance. vOTUs, the number of viral operational taxonomic units; PCs, the number of viral protein clusters; MAGs, the number of metagenome-assembled genomes; EC, electronic conductivity; Ferrous, ferrous iron; Ferric, ferric iron; TOC, total organic carbon; TN, total nitrogen; TP, total phosphorus; AP, available phosphorus; MAT, mean annual temperature; MAP, mean annual precipitation; Dist., distance from the equator; vOTUs abun., the abundance of viruses; and PCs abun., the abundance of viral functions. **b, c** Principal coordinate analysis (PCoA) of viral taxonomic **b** and functional (**c**) structure colored by sampling sites. **d, e** Distance-decay relationships (DDRs) based on Bray-Curtis similarity (1 - dissimilarity) of viral taxonomic **d** and functional **e** community compositions. The blue line denotes the least-squares linear regression across all spatial scales. Red and purple lines denote separate regressions within samples whose distance ≤ 1 km and within samples whose distance > 1 km, respectively. Color-coded best-fit lines and adjusted $R^2$ values for each DDR are presented. The statistical test used was two-tailed. Source data are provided in the Source Data file.

with higher MAP and lower prokaryotic richness. The other direct drivers of viral taxonomic and functional abundance were ferric iron ($r = 0.23$, $P < 0.01$) and pH ($r = 0.20$, $P < 0.05$).

**Virus-host interaction dynamics.** To further resolve potential host effects on viral ecology, we screened the 7,991 high-quality (≥ 50% genome completeness and < 10% contamination) prokaryotic MAGs recovered from the sediment metagenomes for genomic features to link viruses to their putative hosts. As a result, 6,003 viral genomes were linked to 3,404 prokaryotic MAGs. Summarizing these results at the population level revealed virus-host pairs for 3,031 out of the 5,678 vOTUs and 1,488 out of the 2,897 prokaryotic populations (Supplementary Data 4). Most (97%) of the predicted host populations were assigned to 20 prokaryotic phyla, including bacteria belonging to *Proteobacteria* (433 populations), *Actinobacteriota* (193) and *Acidbacteriota* (137) and archaea from the *Thermoplasmatota* (132) (Fig. 4a). The predicted hosts were also affiliated with many poorly characterised phyla, including 14 bacterial populations from the *Dormibacterota*, 13 from *Elusimicrobiota* and 13 from *Eremiobacterota*, and 41 archaeal populations from the *Micrarchaeota*, 17 from *Nanoarchaeota* and 8 from *Thermoproteota*. The abundances of these host phyla were mostly (19 of the 20 phyla) significantly correlated with the total abundance of viruses infecting the same host lineage across the AMD sediments, indicating a high accuracy of our host prediction (Fig. 4a). We also calculated virus-host abundance ratios (VHRs) to assess how virus-host dynamics varied across different hosts. A range of

lineage-specific VHRs (typically > 1) were observed, with the highest average values recorded in *Chloroflexota* (Fig. 4a).

Given the dominance of *Proteobacteria* and *Thermoplasmatota* across the 90 AMD sediments (Supplementary Fig. 3), we examined their virus-host abundance dynamics in detail. The VHRs were significantly higher in *Proteobacteria* than in *Thermoplasmatota* (Supplementary Fig. 4a). We contrasted the abundance between the two phyla across the 90 sediments, and found that *Proteobacteria* and *Thermoplasmatota* showed distinct dynamics in both total abundance and predicted host abundance. The abundance of *Proteobacteria* increased firstly and then decreased along the elevated prokaryotic abundance, while the abundance of *Thermoplasmatota* consistently and substantially increased. These abundance patterns were similar to those of their associated viruses (Fig. 4b). However, the *Thermoplasmatota*-associated viruses showed a weaker increase in abundance compared with their hosts (Fig. 4b). As a result, we found that the total abundance of viruses peaked at intermediate prokaryotic abundance (Fig. 4c).

We next investigated whether prokaryotic hosts might affect viral life strategies and virus-host interaction structure. A deep learning approach was applied to distinguish virulent and temperate viral populations in our data (Supplementary Data 3)[31]. Results showed that the relative abundance of virulent viruses increased while the relative abundance of temperate viruses decreased significantly as the prokaryotic abundance increased, suggesting that virulent life strategies became more prevalent in sediment communities with higher prokaryotic abundance

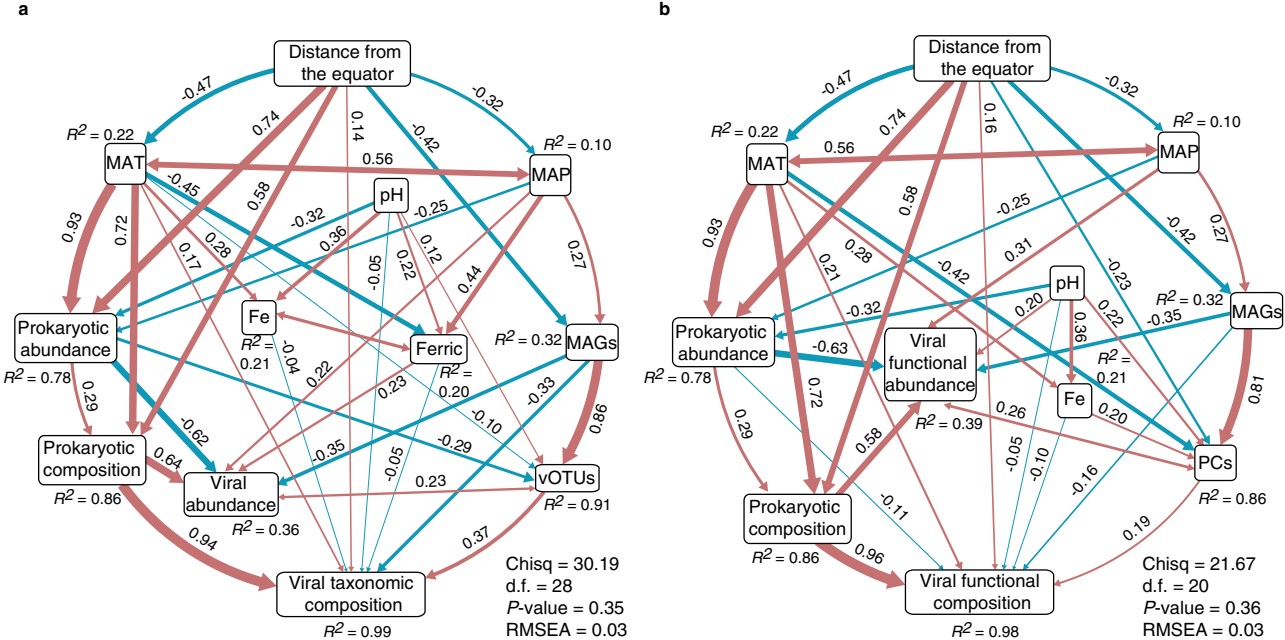

**Fig. 3 Drivers of viral taxonomic and functional diversity and compositions.** Path diagram for SEM showing only significant direct and indirect effects of biotic and abiotic variables on viral taxonomic **a** and functional **b** diversity and compositions. Composition is represented by the PC1 from the Bray-Curtis dissimilarity-based principal coordinate analyses. Numbers adjacent to the arrows are standardised path coefficients ($r$), analogous to relative regression weights and indicative of the effect size of the relationship. Blue and red arrows represent significant ($P < 0.05$) positive and negative pathways, respectively. Double-headed arrows indicate covariance between variables, single-headed arrows indicate a one way directed relationship. $R^2$ represents the proportion of variance explained for every dependent variable in the model. The fit of models was evaluated using one-tailed Chi-squared test and root mean square error of approximation (RMSEA). vOTUs, the number of viral operational taxonomic units; PCs, the number of viral protein clusters; Ferric, ferric iron; MAT, mean annual temperature; MAP, mean annual precipitation; and MAGs, the number of metagenome-assembled genomes. Source data are provided in the Source Data file.

(Fig. 4d). Concomitantly, significant (Wilcoxon $t$-test, $P < 0.001$) higher virulent/temperate abundance ratios were observed in *Thermoplasmatota*-associated viruses than in *Proteobacteria*-associated viruses (Supplementary Fig. 4b). When averaged at the host phylum level, lineage-specific host range (number of host populations for each viral population) and viral range (number of viral populations for each host population) were highest in *Thermoplasmatota* and *Proteobacteria*, respectively. Besides, the host range significantly increased with the prokaryotic richness (Pearson's $r = 0.45$, $P < 0.05$), and the viral range significantly increased with the viral richness (Pearson's $r = 0.86$, $P < 0.001$) across the host phyla (Fig. 5a). Further, increased prokaryotic richness and viral richness were associated with significant decline in modularity (Fig. 5b, d) and significant increase in nestedness of virus-host bipartite sub-networks across the sediment samples (Fig. 5c, e).

**Case study of viral AMGs.** To further elucidate virus-host interactions, we analysed viral AMGs to assess whether abiotic factors impact viral functions, which in turn affect host metabolism and sediment biogeochemistry. We focused on phosphorus (P) metabolism-related genes because of their putative roles in response to P deficiency in AMD environments[32,33]. We identified 75 viral genes annotated as phosphate starvation-inducible protein (*phoH*)[34], which belongs to the COG number of 4QCHF and COG0172 (Fig. 6a and Supplementary Data 5). To further explore the origin of these predicted viral *phoH* genes, 111 homologs from eggNOG database (v5.0.0) and 114 homologs from the recovered MAGs were recruited and combined to build a phylogenetic tree (Fig. 6a and Supplementary Data 6). The result showed that the *phoH* genes were widely distributed in both prokaryotes and viruses and clustered phylogenetically. Further

examination of the recovered *phoH* genes showed that genes assigned as 4QCHF were mostly clustered with their counterparts from viruses and *Bacteroidota*, while genes assigned as COG0172 were mostly affiliated with homologs from *Proteobacteria* and *Patescibacteria*. Interestingly, significant increase in the total abundance of the *phoH* genes was observed with decreasing concentrations of total P (TP) and available P (AP) in the sediments, suggesting that the viral *phoH* genes might be induced under P starvation in AMD sediments (Fig. 6b).

In addition, we assembled a provirus genome encoding the first three genes of the *phn* operon - *phnCDE*, which also belongs to the *pho* regulon and comprises a binding protein-dependent transporter involved in the uptake of P in the form of phosphonate (Fig. 6c and Supplementary Data 5)[35]. This provirus genome covered 72% of the whole fragment that was 'co-binned' with a host population genome (FK3.bin20) classified as *Burkholderiales* of *Gammaproteobacteria* (Supplementary Data 5). Meanwhile, 11 additional *Burkholderiales* populations were predicted as hosts of the provirus based on BLASTn of genomic content, as evidenced by the significant positive correlation between the abundance of provirus and these *Burkholderiales* populations (Supplementary Fig. 5). Furthermore, phylogenetic analyses indicated that the *phnCDE* genes identified in the provirus were affiliated with homologous genes from *Burkholderiales* spp. in eggNOG v5.0.0 database, implying a potential origin of these viral functional genes (Supplementary Fig. 6 and Supplementary Data 6).

## Discussion

Recent metagenomic and viromic surveys have uncovered an unprecedented diversity of viruses in both aquatic and terrestrial environments[12]. Fully accessing viral biodiversity is important for

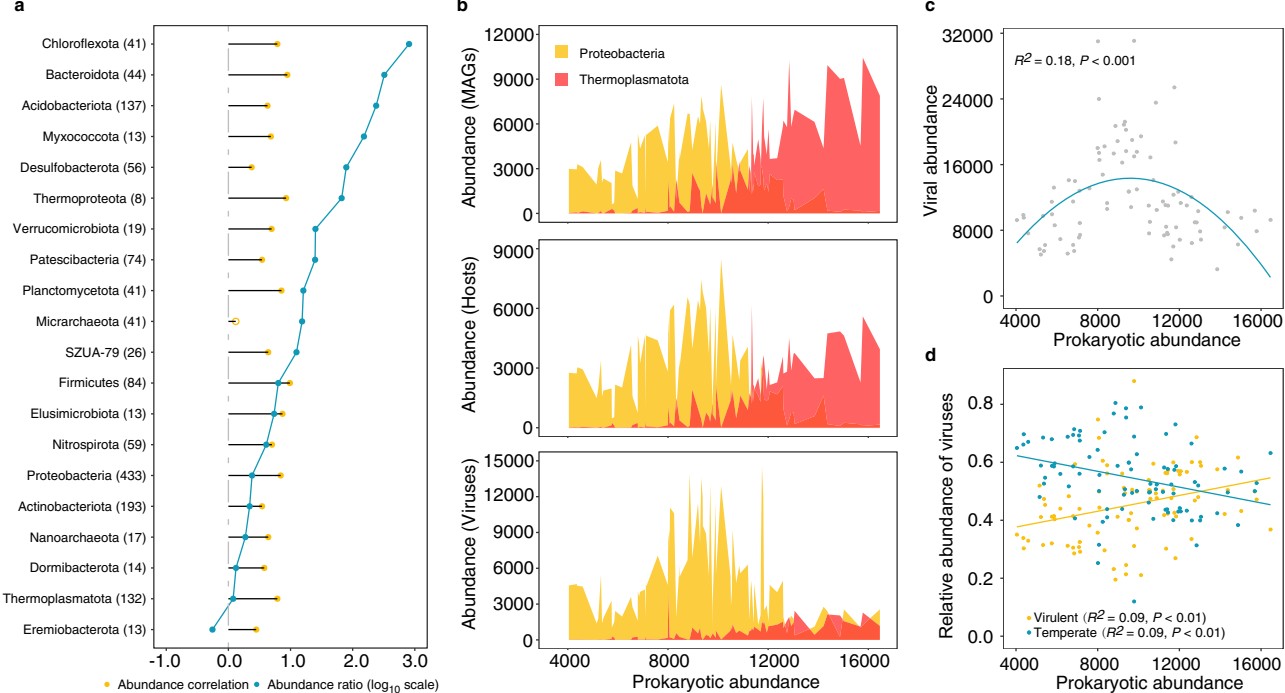

**Fig. 4 Virus-host abundance patterns. a** The blue and orange dots indicate lineage-specific virus-host abundance ratios (VHRs, $\log_{10}$ scale) and virus-host abundance correlation coefficients, respectively. The solid orange dot represents significant ($P < 0.05$, two-tailed test) Pearson's correlations while the hollow orange dots represent nonsignificant correlations. The Pearson's correlations were calculated across samples in which host phyla and associated viruses were both detected. Numbers in the brackets indicate numbers of predicted host populations. **b** Top panel: total abundance of *Proteobacteria* and *Thermoplasmatota*; middle panel: total abundance of *Proteobacteria* hosts and *Thermoplasmatota* hosts; bottom panel: total abundance of *Proteobacteria*-associated viruses and *Thermoplasmatota*-associated viruses. **c** Relationships between viral abundance and prokaryotic abundance. The blue line indicate best-fitting polynomial functions. Adjusted $R^2$ value for this plot is presented. The statistical test used was two-tailed. **d** Relationships between prokaryotic abundance and summed relative abundances of temperate and virulent viruses. Color-coded best-fit lines and adjusted $R^2$ values for each linear regression are presented. The statistical test used was two-tailed. Source data are provided in the Source Data file.

the study of biogeographic patterns but represents a major challenge especially for soil and sediments, where viruses are typically diverse and abundant[29,36]. To bypass this hurdle, we adopted a total metagenome approach to uncover viral taxonomic and functional diversity in AMD sediments and generated a large number of viral genomes and genes. It should be noted, however, that a recent study showed that viromes outperformed metagenomes in recovering viral contigs especially the rare taxa from agricultural soils, indicating the limitation of using metagenomes alone to explore viral communities in complex environmental samples[37]. Thus, a virome-based approach would likely capture more viral populations in our AMD sediments.

Annotation through the reticulate method revealed that a vast majority of our predicted viral genomes could not be taxonomically classified (Fig. 1d), highlighting the uniqueness of viral populations unearthed in the current study. Such a low annotation rate is largely attributable to the absence of complete genomes of viral isolates from AMD and associated environments. This finding suggests that, despite extensive meta-omics analyses of the prokaryotic communities residing the AMD model system[15], our knowledge of the viral biodiversity therein is unbalancedly very limited[20,21,38,39]. Nearly one third of the predicted viral proteins could be annotated by eggNOG v5.0.0 database[30], and they were mostly assigned to known functions that are pivotal for the survival and proliferation of viruses. These metabolic functions have previously been found over-represented in viral assemblages in other habitats[40,41], indicating a universal distribution of viral core genes while there is also evidence of adaptation of certain viral functions to specific environments[42].

The viral taxonomic and functional richness in our study follows the latitudinal diversity gradient paradigm that suggests higher biodiversity in the tropics with a decrease toward the poles (Fig. 2a). While in general agreement with the diversity patterns of other domains of life[43,44], more samples from a wider range of latitudes should be analysed to verify this result. The overall effect of latitude on viral taxonomic and functional richness in the AMD sediments may be primarily attributable to the variations in prokaryotic richness (Fig. 3). However, the role of other factors, in particular pH and MAT, in directly shaping the number of viral populations and functions should not be overlooked. The mechanism explaining the influence of pH and MAT remains unknown, but decreased pH and increased MAT not only exert impacts on prokaryotes and consequently alter the indigenous viral assemblies, but also may increase the fitness cost of viruses persisting in the environment.

Our analyses identified ferric iron concentration as the most important environmental factor governing viral abundance in the AMD sediments (Fig. 2a and Fig. 3). The Ferrojan horse hypothesis has depicted that phages with their tail fibers incorporated with iron ions may effectively infect hosts through competing with siderophore-bound iron for uptake receptors[45]. Therefore, non-Ferrojan viruses would have a fitness advantage in iron-replete conditions[46]. Thus, the iron-rich AMD sediments subsequently may favor the survival and enrichment of non-Ferrojan viruses, contributing to the variation of viral abundance observed in the current study. Another possibility would be potential adsorption of viral particles on iron-bearing minerals precipitated from water phase to the sediments as previous

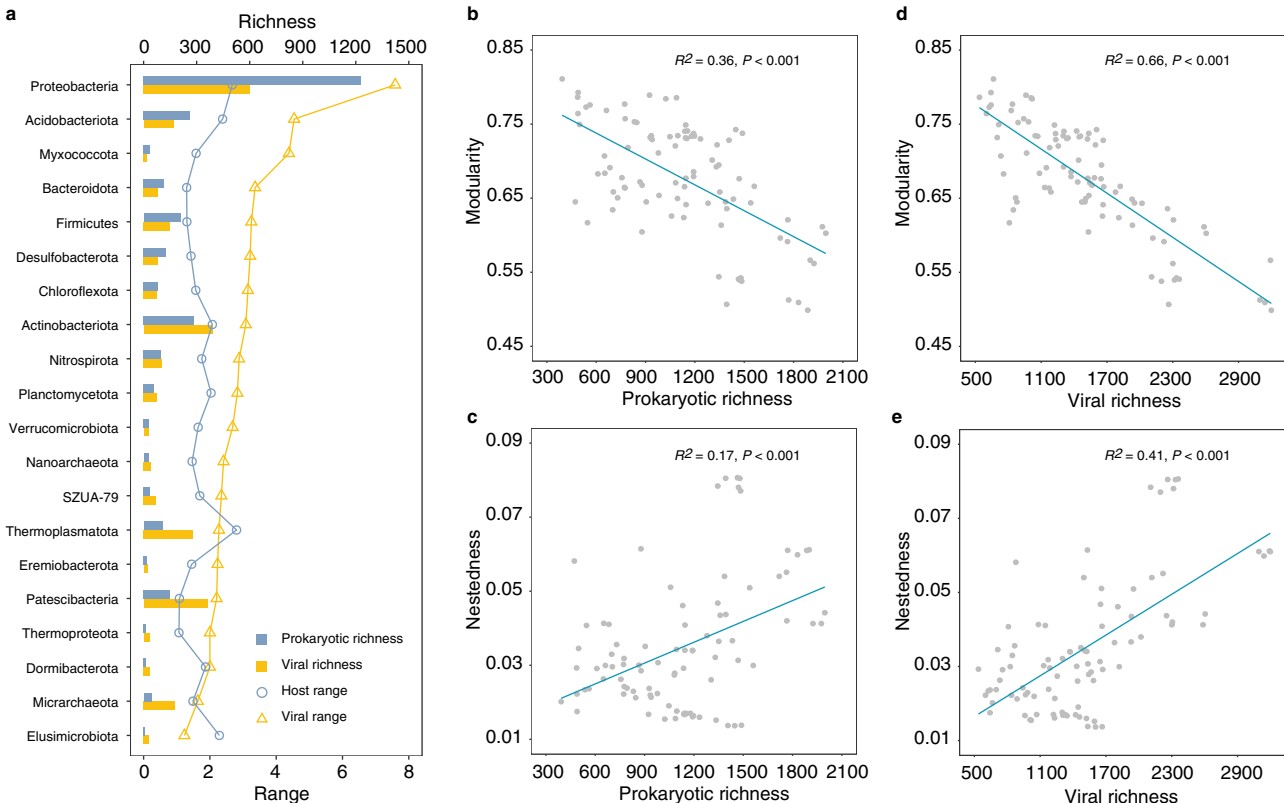

**Fig. 5 Viral-host interaction structure across host lineages and sediment samples.** The prokaryotic richness and viral richness were estimated as the number of prokaryotic MAGs and vOTUs, respectively, within different phyla (**a**) or in sediment samples **b**–**e**. **a** Bar graphs showed the prokaryotic richness (blue) and viral richness (orange), while the line charts indicated host range (circle) and viral range (triangle) by host lineage. The modularity and nestedness in each sample was calculated based on the sub-networks derived from the overall virus-host interaction networks by preserving viral and host populations presented in each sample. The blue lines denote linear regression relationships between prokaryotic richness or viral richness, and modularity (**b**, **d**) or nestedness (**c**, **e**) of sub-networks across the 90 sediment samples. The adjusted $R^2$ values for each linear regression are presented. The statistical test used was two-tailed. Source data are provided in the Source Data file.

investigations have documented strong relationships between viral abundance and mineral saturation indices[47,48]. A similar scenario (i.e., the attachment of viruses on particles and then co-precipitation to the seafloor) has been demonstrated in the marine environment[49]. While being mineral attached may make these viruses inactive, they could subsequently be released with increased pH since minerals with higher isoelectric point tend to be a better adsorbent of viruses[48].

The biogeographic pattern that community similarity decreases with increasing geographical distance has been observed in both prokaryotic and microbial eukaryotic communities[50,51]. Our results extend this pattern to the viral world, revealing a scale-dependent distance-decay distribution of viral taxonomic and functional composition (Fig. 2d, e). Meanwhile, SEM model indicated that MAT, MAP, distance from the equator, and pH were most important in shaping prokaryotic assemblages, which was further the major driver of viral taxonomic and functional composition. This contrasts results from our previous biogeography survey of prokaryotes in AMD solutions where pH was the strongest predictor of microbial community[52], but is consistent with the patterns in marine viruses in that viral communities are influenced by temperature and latitude[13,14]. Furthermore, our data suggest that the distribution of viral populations and functions is unlikely to be primarily affected by environmental variables and geographic distance, but rather by their host compositions. While the strong influence of prokaryotes on viral communities have also been observed in previous studies[19,29], which could be partly attributable to the parasitic lifestyle of viruses, it might also reflect

potential methodological limitations that recovered viral genomes from bulk metagenomes biased toward intracellular viruses and thus should be interpreted with caution[12].

The tight couplings between viral taxonomic and functional composition and prokaryotes were further corroborated by our host prediction analysis, which described numerous virus-host interactions at the population level. Using the predicted virus-host linkages, we demonstrated that almost all viruses exhibited parallel variations in abundance with their hosts (Fig. 4a), which was consistent with genuine virus-host pairs. Notably, total viral abundance was better described as a nonlinear, polynomial function of prokaryotic abundance. This pattern is probably due to the different VHRs between the two dominant phyla: a decrease in the abundance of *Proteobacteria* created niche occupancy for *Thermoplasmatota* to fill, whereas the significantly lower VHRs in *Thermoplasmatota* might result in the observed trend of shallower increase in the abundance of *Thermoplasmatota*-associated viruses (Fig. 4b). Meanwhile, the decrease in viral abundance at higher prokaryotic abundance is unlikely a result of switching of viral life strategies from virulent to temperate, since virulent viruses were more abundant in *Thermoplasmatota*-dominated samples (Fig. 4d). Additionally, the specialisation or generalization of virus-host interactions are subjected to the host group, as indicated by the lineage-specific host range and viral range (Fig. 5a). Furthermore, the prokaryotic and viral richness-related modularity and nestedness supports experimental models that show how the increase of host or viral diversity can select for generalised over specialised phages[53,54].

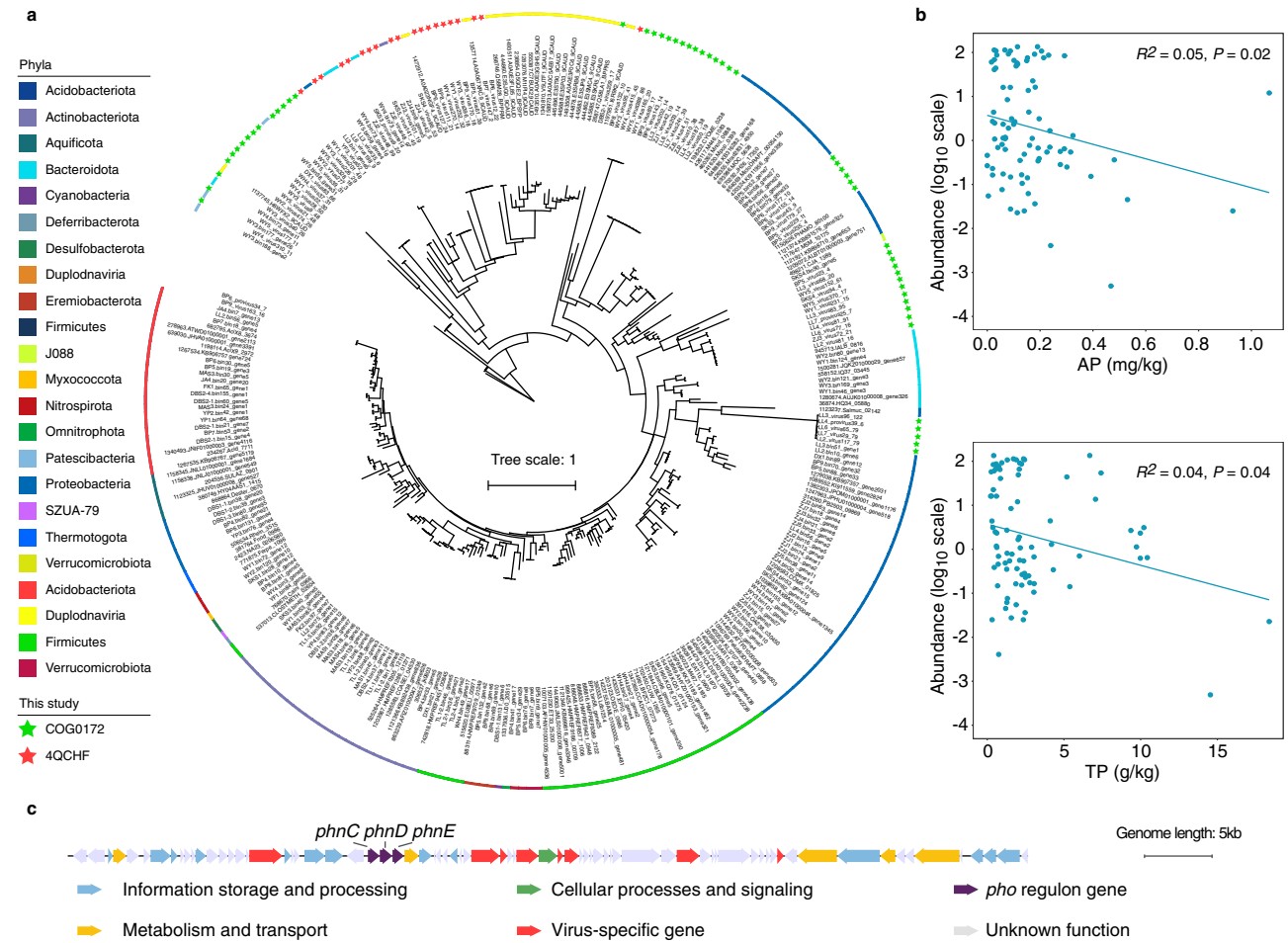

**Fig. 6 Genomic analyses of viral phosphorus (P) metabolism-related genes. a** Maximum-likelihood phylogenetic tree with *phoH* genes from the AMD sediments (indicated by stars) compared to homologs found in eggNOG v5.0.0 database and the host proteins colored by different phyla (the outer color ring). **b** Linear regression relationships between the total abundance of viral *phoH* genes and the concentrations of total P (TP) and available P (AP). The statistical test used was two-tailed. **c** Genome map of a latent provirus genome containing *phnCDE* genes annotated by eggNOG v5.0.0 database. Genes related to information storage and processing are shown in blue; genes related to metabolism and transport are shown in yellow; genes related to cellular processes and signaling are shown in green; virus-specific genes are in red; *phnCDE* genes are in purple; and unknown genes are in grey. Detailed function descriptions of the nine viral scaffolds are listed in Supplementary Data 5. Source data are provided in the Source Data file.

Thus far, very limited information is available for viral AMGs in extreme AMD environments[21]. Our study identified a number of *pho* regulon genes (i.e., *phoH* and *phnCDE*) in the predicted viral genomes (Fig. 6). This suggests frequent horizontal gene transfers (HGTs) of these different types of P metabolism-related genes, which was further supported by the phylogenies of the *phoH* and *phnCDE* genes, as well as previous reports of pho regular genes in viral genomes[55,56]. That none of the viral *phnCDE* genes were affiliated with homologs from the prokaryotic MAGs recovered from the AMD sediments may be a result of mutation events occurred on them. As AMD and associated environments are often oligotrophic, the identified P metabolism-related genes may provide the viruses with the ability to supplement or sustain P assimilation in their hosts, indicating an important adaptation in AMD environments. The observed negative correlations between total abundance of the *phoH* genes and concentrations of TP and AP supported this assumption. It should be noted, however, that the roles and relative importance of phage-encoded *phoH* genes in the P cycle have not been fully resolved[57–59]. Divergent functions such as RNA modification and lipid metabolism have also been documented for these genes[60]. On the other hand, *phoH* has been developed as a novel biomarker for assessing phage diversity in the environment[56]. The identification of *phoH* genes in our AMD sediments provides evidence for the wide distribution of these viral AMGs in different habitats including extreme environments.

Our study contributes to the understanding of viral biogeography by providing an initial view of the community patterns and ecological constraints of viruses populating an extreme environment. Our data suggest that the dynamics of viral populations and functions are subjected to their hosts, and also directly or indirectly correlated with other environmental and geographical variables. Extensive prokaryotic genome recovery from the metagenomic data set further refines our knowledge of how host abundance and diversity may affect virus-host interplays from the point of VHRs and interaction structure, respectively. Future efforts are needed to resolve the mechanisms shaping the viral biogeographic patterns observed in the AMD model system, and to examine whether such findings are relevant to other types of extreme environments on the planet.

## Methods

**Sample collection.** AMD sediments were collected from 18 mine sites in six provinces across Southern China (22.96°−31.68°N, 105.73°−118.63°E) from August to October in 2017[24]. These samples (10 for each site) represent a wide range of mineralogy and environmental conditions. Samples were collected using a shovel from the top 10 cm of AMD sediments either at the center or at ~1 m from

the edge of AMD ponds depending on the safety and size of the features at each mine site. The samples were sealed in 50 mL sterile tubes, kept in an icebox and transported to the laboratory, where they were stored at 4 °C and processed within 24 h. Each sediment was well mixed and divided into two fractions: one fraction for DNA extraction (subsequently stored at −80 °C) and the other for physicochemical measurements (air-dried)[24].

**Environmental measurements.** Geochemical parameters were determined with standard methods[24]. Specially, air-dried subsamples were ground and passed through 20-mesh and 100-mesh sieves, and stored at ambient temperature until use. Total organic carbon (TOC) (TOC-VCPH; Shimadzu, Columbia, MD), total nitrogen (TN) and TP (SmartChem; Westco Scientific Instruments Inc., Brookfield, CT) were analysed with standard methods (0.2 g each). AP was determined colorimetrically by the molybdenum blue method at 700 nm wavelength[61] (5.0 g of subsamples). For measuring pH and EC, 4.0 g of sediments was mixed with 10 mL of deionised water (1:2.5 (w/v)) and the supernatant was then measured using a pH meter and an EC meter. The concentrations of HCl-extractable ferrous iron ($Fe^{2+}$) and ferric iron ($Fe^{3+}$) were determined by UV colorimetric assay with 1, 10-phenanthroline method at 530 nm wavelength (1.0 g of subsamples)[62], and sulphate ($SO_4^{2-}$) was measured by a $BaSO_4$-based turbidimetric method (2.0 g of subsamples)[63]. Total concentrations of heavy metals (including Pb, Zn, Cu, Cd, Fe, and Mn) were determined by inductively coupled plasma optical emission spectrometry (ICP-OES; Optima 2100DV, PerkinElmer, Wellesley, MA) after digestion of 0.2 g sediments with an $HNO_3$/HCl mixture (1:3 (v/v)). Estimates of the MAT and MAP were obtained from the WorldClim2 database (www.worldclim.org).

**DNA extraction and metagenomic sequencing.** Total DNA was extracted from 10 g of each sediment which was pretreated with 30 mL solution containing 0.1 mol/L ethylene diamine tetraacetic acid (EDTA), 0.1 mol/L Tris (pH 8.0), 1.5 mol/L NaCl, and 0.1 mol/L $NaH_2PO_4$ and $Na_2HPO_4$ prior to the employment of the FastDNA Spin Kit (MP Biomedicals, Irvine, CA)[24,64]. Extracted DNA was purified using the QIAquick Gel Extraction Kit (Qiagen, Chatsworth, CA). Finally, a total of 90 samples (with the other samples being discarded due to their low DNA yield/quality) were used for library preparation with NEBNext Ultra II DNA Prep Kit (New England Biolabs, MA) and sequenced from both ends with MiSeq Reagent Kit v3 on an Illumina MiSeq platform (150 bp, paired end reads). This generated totally ~7 Tb metagenomic raw reads data.

**Processing of metagenomic sequence data.** Metagenomic reads were quality filtered and trimmed using in-house Perl scripts. A trim quality threshold of 30 was used and reads containing more than five 'N's were discarded. All quality-controlled reads from a sediment sample were assembled using SPAdes v3.14.1 and kmers of 21, 33, 55, 77, 99, 127 under the '--meta' mode[65]. Genes were predicted by Prodigal 2.6.3 with the parameters as '-p meta -g 11 -f gff -q -m'[66]. For functional annotation, the protein-coding sequences were separately compared against the Pfam v33.1[67], Kyoto Encyclopedia of Genes and Genomes (KEGG) database[68], Non-supervised Orthologous Groups (eggNOG v5.0.0)[30], and Virus Orthologous Group (VOG, http://vogdb.org, Accessed 5 Oct. 2021) with a threshold of 50 for bit score and $10^{-5}$ for E-value. Annotations with the lowest E-value in each database were then selected as the best hits for the proteins.

**Identification and clustering of viral genomes.** Three methods were employed separately to identify viral genomes in the metagenomic assemblies: (1) viral protein families[25], (2) VirSorter v1.0.6 software[26], and (3) CheckV v0.6.0 software[27]. Specifically, viral protein families were downloaded from the Integrated Microbial Genomes with Microbiome (IMG/M) system and used as bait to screen the proteins of metagenomic contigs longer than 10 kb (hmmsearch v3.3.2, threshold of $10^{-5}$ for E-value)[69]. Contigs with five or more viral protein families were collected and then filtered based on the number of genes covered with Pfams and KO terms[25]. Meanwhile, VirSorter (run with default parameters using the 'virome' database) was also used to recover viral contigs longer than 10 kb and those identified as categories 1 and 2 were retained and curated, as described previously[70]. Additionally, prophages identified as VirSorter categories 4 and 5 were processed with CheckV 'contamination' program to identify and remove host contaminations[27]. Finally, viral genomes predicted by the three methods were pooled. All predicted viral genomes originating from eukaryotic viruses based on a BLAST affiliation of the genes to the NCBI RefseqVirus database (ftp://ftp.ncbi.nlm.nih.gov/refseq/release/viral, Accessed 20 July. 2020) were removed[71]. Besides, predicted viral genomes with no genes displaying a best BLAST hit to prokaryotic viruses were also excluded.

The identified viral genomes were clustered into vOTUs using the parameters 95% average nucleotide identity (ANI) and 85% alignment fraction of the smallest scaffolds based on the scripts (https://bitbucket.org/berkeleylab/checkv/src/master/) provided in CheckV[27]. Representative viral population genomes were detected with DeePhage v1.0 to distinguish life strategies (virulent or temperate)[31]. Genes of all identified viral genomes were predicted by Prodigal 2.6.3 (with the parameters set as '-p meta -g 11 -f gff -q -m')[66], and clustered by using cd-hit (-n 4 -d 0 -g 1; 60% identity and 80% coverage)[72]. Reads from each of the 90 sediment metagenomes were mapped to the viral representative genomes and genes using BamM 'make' v1.7.3 (http://ecogenomics.github.io/BamM/) with default parameters, and the coverage of

each sequence was calculated with BamM 'parse' v1.7.3 using the 'tpmean' coverage mode (remove the highest 5% and the lowest 5% coverage regions, minimum nucleotide identity of 95%, minimum aligned length of 75% of each read). The abundance for a given scaffold or gene was computed as the average scaffold or gene coverage divided by the number of reads in a given library and multiplied by the mean value of the number of reads in the 90 libraries. For taxonomic assignment, a gene content-based network analysis was used to taxonomically place the viral representative genomes in the context of known viruses[28]. Briefly, predicted proteins from viral genomes were clustered with predicted proteins from isolate reference viruses (v201) based on an all-versus-all BLASTp search with an E-value of $10^{-3}$, and protein clusters were defined with the Markov clustering algorithm and processed using vConTACT v2.0[28]. Meanwhile, predicted viral proteins were aligned against the NCBI Viral RefSeq v201 database using BLASTp with a threshold of 50 for bit score and $10^{-5}$ for E-value. The LCA algorithm was then used for taxonomic analysis of each viral genome based on the taxonomic rank of annotated proteins[29].

**Recovery of prokaryotic population genomes.** Prokaryotic population genomes were recovered from the 90 sediment metagenome assemblies (excluded free viral genomes) using MetaBAT v2.12.1[73], MaxBin v2.2.2[74], Abawaca v1.00[75], and Concoct v0.4.0[76] with default parameters, considering tetranucleotide frequencies, scaffolds coverage and GC content. The resulting bins were then combined using DASTool v1.1.2[77], and further manually curated to obtain high-quality genomes using RefineM v0.0.24[78]. These genomes were then classified using the genome taxonomy database (GTDB-Tk v1.6.0)[79]. The completeness and contamination of genome bins were assessed using CheckM v1.1.3 with default parameters, except those assigned as *Patescibacteria* which were estimated using a smaller set of markers[80]. Genomes estimated to be ≥ 50% complete and < 10% contaminated were selected to calculate the ANI. Genomes with > 97% ANI over >70% alignment were grouped as a population: the highest quality genome calculated as 'completeness − 4 × contamination' in each population was chosen as the representative[81]. Finally, reads from each of the 90 sediment metagenomes were mapped to the set of dereplicated genomes using BamM v1.7.3 as described above for the viral sequences (Supplementary Data 7).

**Virus–host linkage analyses.** Viral genomes were putatively linked to their hosts in silico[82]. Briefly, these linkages were based on (1) shared genomic content between viral scaffolds and host genomes, (2) prophages identified in host genomes, and (3) sequence similarity between CRISPR-spacers in host genomes and protospacers in viral scaffolds. All viral genomes were compared to the recovered prokaryotic genomes using BLASTn (E-value ≤ $10^{-3}$, bit score ≥ 50, alignment length ≥ 2.5 kb and identity ≥ 70%)[71]. Viral genomes identified as prophages were matched to their corresponding host genomes. CRISPR spacers were recovered from metagenomic scaffolds using metaCRT with default parameters[83]. Extracted spacers were compared to viral scaffolds using BLASTn with thresholds of an E-value ≤ $10^{-10}$ and no mismatches over the whole spacer length[71,84].

**Viral AMGs analyses.** The predicted viral proteins were assigned to eggNOG v5.0.0 database using BLASTp (threshold of 50 for bit score and $10^{-5}$ for E-value)[30]. As a result, 75 viral proteins were assigned as *phoH* genes (4QCHF and COG0172) and three were assigned as *phn* operon (*phnCDE*). These viral proteins were compared to the host proteins and eggNOG v5.0.0 database (BLASTp, threshold of 50 for bit score and $10^{-3}$ for E-value) to recruit relevant sequences (up to 5 for each viral AMG sequence)[71]. Each set of viral AMGs were then aligned with Muscle v3.8.31 and filtered by TrimAL v1.4.rev22 to remove columns comprised of more than 95% gaps[85,86]. Finally, phylogenetic trees were constructed using iqtree2 with the parameters set as '-mem 100GB -T 20 -m MFP -B 1000 --bnni', and visualized and formatted in the Interactive Tree of Life online interface using the Newick file with the best tree topology[87,88].

**Statistical analyses.** Statistical analyses were implemented with various packages within the statistical program R v4.0.3[89]. Biotic and abiotic matrices were standardised using 'decostand' function in vegan v2.5–5 with methods of 'Hellinger' and 'Standardize', respectively[90]. Bray–Curtis dissimilarity was used to show distances for prokaryotic and viral community structure and function profiles, whereas Euclidean distances were calculated using environmental variables (vegan v2.5–5)[90]. Pearson correlations were performed using 'rcorr' function (999 permutations) in Hmisc v4.2-0 to assess the relationships between the richness and abundances of viral populations and functions, prokaryotes and environmental variables in all samples[91]. Mantel tests were performed to reveal the correlations between the dissimilarity matrices (vegan v2.5-5)[90]. In all correlation analyses, $P$ values were adjusted for multiple testing using the Benjamini and Hochberg false discovery rate controlling procedure (stats v4.0.3)[92]. To understand how local spatial organisation of the viral communities varies within and across different AMD sites, PCoA (utilizing the Bray-Curtis dissimilarity metric), which allows dimensionality reduction, was used (vegan v2.5-5)[90]. The rate of the DDRs was calculated as the slope of a linear least squares regression on the relationship between log10-transformed geographical distance versus viral taxonomic and functional community composition similarity. SEM was used to tease apart the direct and indirect relationships among environmental and geographical variables,

prokaryotic community composition, and viral taxonomic and functional composition (lavaan v2.1.2)[93]. Community composition was represented by PCoA PC1 based on the Bray-Curtis dissimilarity metric. Priori models were first constructed, considering all theoretical or empirical mechanisms whereby abiotic and biotic factors influence viral taxonomic and functional diversity, abundance and structure (Supplementary Fig. 2). The priori models were then optimized until attaining the final models. A Chi-squared test and the RMSEA were used to evaluate the fit of models. Sub-networks for virus-host interactions in each sediment sample were also generated from meta-networks by preserving viral or prokaryotic populations presented in the sample. The modularity and nestedness values for each sub-network were computed with 'Brim' and 'NODF' algorithm in MATLAB BiMat package with 1000 permutions[94]. The Shapiro-Wilk test and Bartlett's test were performed to check for normality and equal variance between groups[92]. Statistical significance of differences was then determined using non-parametric Wilcoxon t-test (unpaired)[92].

**Reporting summary**. Further information on research design is available in the Nature Research Reporting Summary linked to this article.

## Data availability

Raw reads of metagenomes and all assembled prokaryotic population genomes have been deposited in NCBI BioProject database under accession code PRJNA666025. Short Reads Archive accession numbers for individual reads are listed in Supplementary Data 8. Biosample accession numbers for individual prokaryotic genomes are listed in Supplementary Data 9. Assembled viral genomes are available from the NCBI BioProject database under accession code PRJNA648034. eggNOG database is available at http://eggnog5.embl.de/download/eggnog_5.0. NCBI viral RefSeq database is available at https://ftp.ncbi.nlm.nih.gov/refseq/release. WorldClim database is available at https://www.worldclim.org/data/worldclim21.html. Source data are provided with this paper.

## Code availability

The in-house Perl scripts, R scripts, Matlab scripts, and relevant data used to generate figures of this study are provided with this paper and publicly available on GitHub at https://github.com/eco-gaoshaom/viral-biogeography (https://doi.org/10.5281/zenodo.6374561).

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

## Acknowledgements

This work was supported by the National Natural Science Foundation of China to L.N.H. (nos. 31870111 and 31570500) and to W.S.S. (no. 41830318), as well as by the Natural Science Foundation of Guangdong Province to L.N.H. (no. 2021A1515012468).

## Author contributions

S.M.G., L.N.H., and W.S.S. designed the experiments. S.M.G., H.X.A., and J.Z. conducted the experiments and collected the data. S.M.G., Z.H.L. and H.C. analysed the data. S.M.G. and L.N.H. wrote the initial draft of the manuscript while D.P.-E., J.L.L. provided substantial feedback.

## Competing interests

The authors declare no competing interests.
