## [Peer Review File · Nature Communications]

Reviewers' Comments:

Reviewer #1:

Remarks to the Author:

In recent years it has been possible to begin to explore viral populations in different environments; in general, the microbiological study has been biased towards prokaryotes, leaving aside the effect that viral populations have on different communities and ecosystems. This work is very relevant because it allows having a broader panorama of the effects of the particular physicochemical conditions as well as the climatic ones from extensive sampling of the AMD of southern China, evaluating the interactions that viruses have with their hosts thus as the dynamics that arise in this type of ecosystems in particular.

As you well say in the discussion, it must be taken with caution to say how much the hosts influence the distribution of viruses, since although viruses cannot replicate in their absence, there is still a large number of viral sequences of which we do not yet know its role in the communities' dynamics and, on the other hand, host specificity has not been explored in depth.

Reviewer #2:

Remarks to the Author:

Gao et al. describe host virus linkages over a wide range of acid mine drainage sediments and focus on a case study of phosphorus acquisition auxiliary metabolic genes.

This is a staggering dataset which by itself would be a great addition to the field. Moreover, the paper is very well written; it is clear, concise and cohesive. I really enjoyed reading it.

All in all this work is valuable, promotes the field, the methodology is sound and i can imagine many more papers coming out of it.

Most of my comments are minor and there are a few points in the discussion that i feel are overreaching, but otherwise my biggest concern that the metagenomic reads and MAGs are not publicly available.

Specific comments:

Line 78: Comics? I assume you meant omics

Line 79: change strike to strive

Line 143: in support of this

Line 146: why did you choose a linear correlation? What i see is two unstructured clusters which lead to the same conclusion that there is a cutoff at 1 km for dissimilarity.

Figure 2e: in the legend the red line looks yellow

Lines 183-184: i would mention that populations were achieved by dereplicating MAGs.

Line 194: it's even more impressive to me that 13/20 have an $R^2 > 50$.

Line 219: i would rephrase to say they were placed in a virus specific clade.

Discussion: there are some seminal papers I don't see cited here, like Tyson and Banfield and Sun et al. also from the Banfield group.

Line 249-251: our knowledge on any environmental system is incomplete. This is a moot point.

Line 251-257: the genes you are describing are the core genes that a virus cannot survive without. Of course they are universally distributed in viruses. Once again this is a moot point and should be replaced with a stronger conclusion.

Line 271: i doubt iron is limiting in these systems, but i was wondering if you were aware of the Ferrojan horse theory on vital attachment to iron receptors. It could be a cool link here.

Line 275-281: does the pH in the sediment then release the viruses from the particles? Because otherwise being mineral attached would make them inactive.

Line 302-303: again this is a moot point. Of course viruses depend on their hosts for replication. What's interesting is discussing why the correlations mostly don't explain a huge part of the variability. It probably has to do with the broad taxonomic level of this analysis (e.g. Proteobacteria is a huge and incredibly diverse phylum), and probably with attachment to minerals.

Line 307-308: could you be more specific about these proposed phylum level infection mechanisms? That seems overreaching to me.

309: you have no evidence of active viral replication. To do that you need either a time series or

isotope studies. You have evidence of historical infections. Remove this sentence.

Line 311: there are so many extreme environments not mentioned here, like the deep ocean, deserts and volcanic rock to name a few. I don't appreciate this attempt to spin the conclusions into a bigger frame, especially when there's no need of it. You have such a great dataset that spans AMD environments over space, pH etc. Also this sentence feels disconnected from the rest of the paragraph.

Line 334: environments

Line 434-447: this should be a separate section on MAG generation.

Reviewer #3:

Remarks to the Author:

The manuscript prepared by Gao et al. which was submitted to Nature Communications outlines an observational study wherein the authors examined the viral communities (and their hosts) extracted from acid mine drainage (AMD) sediments collected from mine sites around China. They explore host dynamics, virus dynamics and phosphorus functional genes to describe biogeography trends and community functions. It is not entirely clear to me why they used amplicons alongside the soil metagenome, though I presume it assisted in separating putative viruses from their hosts; not something we do in my own lab, but an interesting approach to the problem. The methods and analyses are sound throughout, and the narrative structure, writing, etc. is suitable for the subject matter. I have highlighted several places within the manuscript that would benefit from some revisions – sentences that are missing a word, or with poor sentence structure, or with odd word choices. I also recommend a slight expansion of the methods, particularly on the topic of sampling. Please see the file attached for grammatical corrections and further comments.

Reviewer #4:

Remarks to the Author:

In this paper, viral populations and their distributions were characterized in acid mine drainage sediments across China. These analyses came from 90 metagenomes in which MAGs were reconstructed and there was additional 16S rRNA gene amplicon data. Overall, it was enjoyable to read, you can tell the authors put in a large amount of effort, and it is very important to understand viral and microbial communities and their dynamics in anthropogenic environments. This was a lot of work and included great analyses determining ecological drivers of the viral communities based on abiotic and biological factors. For the most part, I was impressed to see extensive work was done to make these analyses top notch (e.g., ASVs over OTUs, metaG and Amplicon work, SILVA over Greengenes database, MAG reconstruction,...). This is also somewhat true for the virus work, except it was sad to see the 10kb threshold that is important for vOTUs not used (literature was cited which highlights the importance of the 10 kb threshold). This is worrisome but can be easily overcome if the analyses could be redone with the smaller contigs removed. But if the conclusions remain the same, then the current data could be kept with the extra work noted. The authors will have trouble making the case for vConTACT2 work, because the genera will look much different. This is a minor part of the paper and could be removed if they do not want to do the re-analyses.

After reading the document a couple times, I realized that there are a lot of results and only some make their way into the discussion section. For example, figure 4. Part A shows some of the linkages (even though the figure says it shows all) and I am not sure what the reader is supposed to get from this. It would be great to know if there are more connections with specific phyla and how that may impact the environment. As it is now, we see dots connected. Part B Shows the abundances of vOTUs and MAGs in each sample. This is great but where is the interpretation? Part C shows what is in part B but now with all the samples combined. From looking at the data, I see some viruses correlate with hosts and some don't. It looks like regression analyses were thrown at each vOTU population connected to a phylum, no matter what it looked like. What are we supposed to get from this? Are some vOTUs suspected to have higher burst sizes, or do lytic infection more? It adds nothing to the paper as is. Part D is another way of showing the data in part b and c, with no interpretation. I could see parts b and d being useful if there was a targeted

story to highlighting specific microbial phyla and how their dynamics with linked vOTUs may impact the environment.

This is also done for figure 5. Here, I was pleased to see in the results a mention that there was "11 additional Burkholderiales populations" that were highlighted, but then no discussion.

I would caution the interpretation of the geography analyses in regard to distance from the equator because the samples come from only southern China and viruses are pulled from traditional metagenomes (as compared to a virome).

Some of the word choice is off and needs to be fixed, but overall the writing and flow are good. Viral populations were defined as vOTUs, but there is switching between writing it out and writing vOTU. After the first instance in which the abbreviation is given, please only use vOTU.

I have attached line specific comments. I think this work can be published here, but it needs major revisions.

Great work and I hope to see another version of this manuscript. Please reach out to me if further clarification is needed on any of the comments.

Best,

Gary Trubl (Trubl1@Ilnl.gov)

Patterns and ecological drivers of viral communities in acid mine drainage sediments

In this paper, viral populations and their distributions were characterized in acid mine drainage sediments across China. These analyses came from 90 metagenomes in which MAGs were reconstructed and there was additional 16S rRNA gene amplicon data. Overall, it was enjoyable to read, you can tell the authors put in a large amount of effort, and it is very important to understand viral and microbial communities and their dynamics in anthropogenic environments. This was a lot of work and included great analyses determining ecological drivers of the viral communities based on abiotic and biological factors. For the most part, I was impressed to see extensive work was done to make these analyses top notch (e.g., ASVs over OTUs, metaG and Amplicon work, SILVA over Greengenes database, MAG reconstruction,...). This is also somewhat true for the virus work, except it was sad to see the 10kb threshold that is important for vOTUs not used (literature was cited which highlights the importance of the 10 kb threshold). This is worrisome but can be easily overcome if the analyses could be redone with the smaller contigs removed. But if the conclusions remain the same, then the current data could be kept with the extra work noted. The authors will have trouble making the case for vConTACT2 work, because the genera will look much different. This is a minor part of the paper and could be removed if they do not want to do the re-analyses.

After reading the document a couple times, I realized that there are a lot of results and only some make their way into the discussion section. For example, figure 4. Part A shows some of the linkages (even though the figure says it shows all) and I am not sure what the reader is supposed to get from this. It would be great to know if there are more connections with specific phyla and how that may impact the environment. As it is now, we see dots connected. Part B Shows the abundances of vOTUs and MAGs in each sample. This is great but where is the interpretation? Part C shows what is in part B but now with all the samples combined. From looking at the data, I see some viruses correlate with hosts and some don't. It looks like regression analyses were thrown at each vOTU population connected to a phylum, no matter what it looked like. What are we supposed to get from this? Are some vOTUs suspected to have higher burst sizes, or do lytic infection more? It adds nothing to the paper as is. Part D is another way of showing the data in part b and c, with no interpretation. I could see parts b and d being useful if there was a targeted story to highlighting specific microbial phyla and how their dynamics with linked vOTUs may impact the environment.

This is also done for figure 5. Here, I was pleased to see in the results a mention that there was "11 additional Burkholderiales populations" that were highlighted, but then no discussion.

I would caution the interpretation of the geography analyses in regard to distance from the equator because the samples come from only southern China and viruses are pulled from traditional metagenomes (as compared to a virome).

Some of the word choice is off and needs to be fixed, but overall the writing and flow are good. Viral populations were defined as vOTUs, but there is switching between writing it out and writing vOTU. After the first instance in which the abbreviation is given, please only use vOTU.

Below are line specific comments in blue.

Abstract

Lines 32-33 “The results demonstrated that host communities manipulate viral taxonomic and functional diversity”

I know what you mean by this and I agree but the word manipulate is poor word choice. What viruses are present, their diversity, and largely their abundance is dictated by host diversity and abundance. Please change manipulate to dictate, influence or a similar verb.

Lines 36 “case analyses”

What does this mean? Do you mean a case as an example from your data? Please clarify and make the writing clearer.

Lines 58-59 “and facilitating horizontal gene transfers (HGTs) via transduction.”

I would remove “via transduction” because they also transfer genes by lysing hosts, making those genes available for uptake (transformation).

Lines 61-62 “While omics approaches have later been applied to explore viral diversity in the environment”

We are talking about metagenomics, metatranscriptomics and etc., so please say meta-omics or meta ‘omics and not omics. Omics is constantly used incorrectly because it alone means genomics, transcriptomics, and more, but not meta analyses.

Line 64 “visual communities”

What do you mean by visual? Detected? Please clarify.

Line 78 “omics approaches”

I think you meant omics, but it should be meta-omics.

Lines 79-81 “Here we strike to address this knowledge gap by utilizing a massive metagenomic data set generated from 90 acid mine drainage (AMD) sediments sampled across Southern China (Fig. 1a).”

Poor word choice with “strike”, please change.

Lines 83-88 “The biogeographic patterns of viruses in our data reveal strong impacts of prokaryotes, as well as both direct and indirect influence of abiotic factors on viral populations and functions. Moreover, extensive reconstruction of host genomes and prediction of virus-host interactions illustrated possible mechanisms for how viral communities are structured with respect to host population dynamics.”

The results and discussion should not be in the introduction, please move the writing or delete. Just lay out background on what has been done and what you did for context.

Results

Line 91 “Overall viral diversity and novelty in the AMD sediments.”

Remove overall

Lines 97-99 “In total, we identified 7,442 potential viral populations (viral operational taxonomic units, vOTUs), which are suggested to approximately represent species-level taxonomy”

This is a lot and very exciting. Please make sure to list the contig cutoff, because in the previous sentence you give a bunch of ranges, so it is unclear. The cited paper gives guidelines of 10kb cut off for vOTUs.

Lines 113-115 “Functional composition of the viral communities was examined by comparing the predicted viral proteins against the eggNOG database (v5.0.0)24.”

This is good but in future work, I suggest using one or more of these: VOGDB (<https://vogdb.org/>), MultiPhATE2 (<https://academic.oup.com/g3journal/article/11/5/jkab074/6178284?login=true>), DRAM-v (<https://academic.oup.com/nar/article/48/16/8883/5884738>), VIBRANT (<https://microbiomejournal.biomedcentral.com/articles/10.1186/s40168-020-00867-0>).

Lines 144-145 “Bray-Curtis similarities”

Please change similarities to dissimilarities. Bray-Curtis dissimilarity is used to quantify the differences in populations and is not the same as similarity. Dissimilarity = 1 - Similarity and can push populations together just because of how different they are to other populations.

Lines 178-182 “Extensive genome reconstruction was performed for the bacteria and archaea present in the sediments to link viral populations to their hosts. Finally, we recovered 7,759 prokaryotic MAGs (> 50% genome completeness and < 10% contamination), of which 3,948 were linked to 8,428 viral genomes.”

CPR biology - particularly host relationships - is very much an emerging field, with lots left to learn. Checkm on default parameters systematically underestimates CPR genome completeness. This is due to the fact that the program is not aware of the lineage specific losses of typical bacterial marker genes and thus the genomes appear to be less complete than they actually might be. The Checkm authors have provided a temporary work around using a smaller set of markers proposed by the Banfield lab: <https://github.com/CoGenomics/CheckM/wiki/Workflows#using-cpr-marker-set>. In short, you might actually have some "high quality" CPR.

I did check the methods and read this “Prokaryotic population genomes were recovered from the 90 sediment samples using MetaBAT v2.12.162, MaxBin v2.2.263, Abawaca v1.0064, and Concoct v0.4.065 with default parameters, considering tetranucleotide frequencies, scaffolds coverage and GC content.”. This is extensive, which is great. Please do look into the CheckM comment and denote how many or high quality.

The next sentence starts with “finally” and this is odd because it is the third sentence and not part of a list.

Lines 184-186 “Most (98%) of the predicted host populations were assigned to 20 prokaryotic phyla each with more than 10 populations matched with their viruses.”

At lot of information is hidden in this sentence and it would be great to add in some more detail (unless there is a strict word limit). Most of the taxa are bacterial; please list # of bacteria phyla and # of archaeal taxa. I assume the latter part means there were 10 different vOTUs linked to each phylum? Expand on this. What percent of vOTUs and MAGs were linked? Was it with more than one method? Were many of the vOTUs linked to multiple MAGs?

Lines 189-191 “We compared the networks with the null bipartite matrices with 95% confidence coefficient to evaluate the statistical significance of the modularity²⁵ and found that all networks displayed modular topology (Fig. 4b).”

For parts a and b in this figure only some of the data is shown and the rest is in different supplementary figure. This is not stated in the text or in the figure 4 legend. There is no explanation why a subset of the data was shown. I understand the lack of space, but you need to be transparent on why this data was selected and provide details of where to find the rest. Most readers do not look into the supplementary work unless there is something specific they are investigating. Please help the reader.

Discussion

Lines 240-243 “To bypass this hurdle, we adopted a total metagenome approach to uncover viral taxonomic and functional diversity in the AMD sediments, and generated saturated number of AMD viral genomes and genes (Fig. 1c).”

I think an “a” is missing between generated and saturated. It is also a little odd to say that a saturated number was generated. Technically we have no idea, but the abundance curves suggest these are well sampled for dsDNA viruses. We do know that metagenomes are bad at capturing viruses and I would add in a caveat in the text that a virome would likely capture more.

As an example, see:

Santos-Medellin, C., Zinke, L.A., Ter Horst, A.M., Gelardi, D.L., Parikh, S.J. and Emerson, J.B., 2021. Viromes outperform total metagenomes in revealing the spatiotemporal patterns of agricultural soil viral communities. *The ISME Journal*, pp.1-15.

Lines 273- “Previous investigations...”

This is great work, but you are missing other possibilities. What about viruses using iron for infection. Please incorporate some of this literature in the discussion:

Bonnain, C., Breitbart, M. and Buck, K.N., 2016. The Ferrojan horse hypothesis: iron-virus interactions in the ocean. *Frontiers in Marine Science*, 3, p.82.

Muratore, D. and Weitz, J.S., 2021. Infect while the iron is scarce: nutrient-explicit phage-bacteria games. *Theoretical Ecology*, pp.1-21.

Lines 308-310 “Additionally, based on lineage-specific VHRs, our results also revealed active viral replication and a possibly top-down control of prokaryotes via viral lysis in the AMD sediments (Fig. 4d).”

Metagenomes can show who is there and the potential, but alone it cannot show activity. Comparing viral and microbial advances from metagenomes does not describe activity, even if it was done over time or across different samples. Please remove this sentence.

If you wanted to determine this in future work, consider adding a metatranscriptome with a metaproteome, stable isotope probing, or BONCAT as a few examples.

Lines 321-323 “The identification of *phoH* genes in our AMD sediments provides evidence for the wide distribution of these viral AMGs in different habitats including extreme environments.”

The authors have done a great job of listing methodological limitations, but it is not done here. I appreciate that more P work beyond just *phoH* was done, but you need to bring dissenting literature in for context. The function of virally encoded *phoH* is not clear, and *phoH* expression in phosphate limited conditions appears to vary between hosts. The *phoH* gene is contentious because there is work showing that it may be involved in P cycling, but other work showing the gene is used in other ways and doesn't impact P cycling. There is no reference here highlighting the other work, which needs to be mentioned even if you spin to say this work will aid in disentangling or shed light on the truth. Below are some other works that I found on the *phoH* gene.

Sullivan et al. (2005): "Although the *phoH* gene is found widely distributed among both eubacteria and archaea [79], including all cyanobacteria, and is known to be induced under phosphate stress in *E. coli* [80], its function has not been experimentally determined. Bioinformatic analyses suggest that these *phoH* genes are part of a multi-gene family with divergent functions from phospholipid metabolism and RNA modification (COG1702 *phoH* genes) to fatty acid beta-oxidation (COG1875 *phoH* genes) [79]."

Zeng and Chisholm (2012) "*phoH* encodes an ATP binding protein with unknown function [15] and is considered a phosphate (*pho*) regulon gene because it is upregulated by P starvation in *E. coli* [16, 17]. Its expression is not upregulated during P starvation in marine cyanobacteria [2], however, suggesting that it may not play the same role as in *E. coli* [18]. Nonetheless, because of its prevalence in T4-like cyanophages [9] and association with the *pho* regulon in *E. coli*, we examined its expression in our experiments. We found that the expression of *phoH* in the phage (Figures 1C and S1B) and host (Figure S1A) was not affected by P starvation, and therefore its role in both host and phage remains a mystery."

Sullivan, M.B., Coleman, M.L., Weigle, P., Rohwer, F. and Chisholm, S.W., 2005. Three *Prochlorococcus* cyanophage genomes: signature features and ecological interpretations. *PLoS Biol*, 3(5), p.e144.

Zeng, Q. and Chisholm, S.W., 2012. Marine viruses exploit their host's two-component regulatory system in response to resource limitation. *Current Biology*, 22(2), pp.124-128.

Warwick-Dugdale, J., Buchholz, H.H., Allen, M.J. and Temperton, B., 2019. Host-hijacking and planktonic piracy: how phages command the microbial high seas. *Virology journal*, 16(1), pp.1-13.

Tetu, S.G., Brahamsha, B., Johnson, D.A., Tai, V., Phillippy, K., Palenik, B. and Paulsen, I.T., 2009. Microarray analysis of phosphate regulation in the marine cyanobacterium *Synechococcus* sp. WH8102. *The ISME journal*, 3(7), pp.835-849.

Lindell, D., Jaffe, J.D., Coleman, M.L., Futschik, M.E., Axmann, I.M., Rector, T., Kettler, G., Sullivan, M.B., Steen, R., Hess, W.R. and Church, G.M., 2007. Genome-wide expression dynamics of a marine virus and host reveal features of co-evolution. *Nature*, 449(7158), pp.83-86.

Methods

Lines 341-343 "Samples were collected in 50 mL sterile tubes, kept in an icebox and transported to the laboratory, where they were stored at 4 °C prior to subsequent analyses."

DNA degrades and microbes are active at 4 °C. The exact storage time needs to be given and if longer than 1-2 days, an explanation needs to be given. If several days, then this needs to be stated in the results section and some in the discussion.

Lines 345-348 “Air-dried subsamples were analysed with standard methods for the determination of TOC (TOC-VCPH; Shimadzu, Columbia, MD), total nitrogen (TN) and total phosphorus (TP) (SmartChem; Westco Scientific Instruments Inc., Brookfield, CT).”

How many grams of sediment was used?

Lines 358-359 “For the 90 samples,”

How much sediment was used?

Lines 413-416 “The identified viral genomes were then clustered into vOTUs using the parameters 95% ANI and 85% alignment fraction of the smallest scaffolds, and the longest viral genome in each vOTU was chosen as the representative viral population genome.”

I am confused why all the correct thresholds from Roux et al. 2019 were used except the 10kb cutoff? I am seeing more of this in the literature without additional genome characterization to make sure contigs were not split or to determine true viral origin. More data is not always better and here it lowers the quality of the work. I would like to see the smaller vOTUs used, unless it can be shown that there are minimal differences.

Roux et al. 2015

“while providing near-perfect identification (>95% Recall and 100% Precision) on contigs of at least 10kb.”

Roux et al. 2019

“Importantly, current methods for automatic virus sequence identification cannot reliably identify short (<10 kb) viral sequences, which should be interpreted with utmost caution.” This applies to the VirSorter version used in this paper.

The 10kb cutoff is critical for diversity analyses and vConTACT generation of genera. “*Some of these approaches require a minimum contig size—for example, contigs ≥ 10 kb for taxonomic classification based on gene content or diversity estimation—and will not be applicable to every genome fragment.”

Jang et al. 2019

vConTACT is not perfect but very good. It is largely limited to RefSeq viruses only. Applying it to vOTUs from metagenomes is questionable because it largely relies on the database size and contigs lengths. I would be okay with some conclusions if there was a 10kb cutoff because that is what vConTACT2 was evaluated on. “To evaluate scalability of our algorithm, we added 15,280 curated viral genomes and large genome fragments (≥ 10 kb) from the Global Ocean Virome (GOV) dataset to our reference network in 10% increments (that is, 0%, 10%, ..., 100% of the total dataset).”

Lines 475-478 “Bray–Curtis distances were used to construct the dissimilarity matrices for prokaryotic and viral community structure and function profiles, whereas Euclidean distances were calculated using environmental variables (vegan package v2.5-5).”

This is great but some minor rewording. Bray–Curtis dissimilarity can be used to identify distance, so something along this line is better “Bray–Curtis dissimilarity was used to construct to show distances for

prokaryotic and viral community structure and function profiles, whereas Euclidean distances were calculated using environmental variables (vegan package v2.5-5).”

Line 490 “utilizing the Bray-Curtis distance”

Change distance to dissimilarity metric

Line 498 “Bray-Curtis distance”

Change distance to dissimilarity metric

Line 510 Data availability

What about the metagenomic reads and the MAG bins?

Figure 2

Throughout the paper part B and C are described as PCoA's but in fact they are NMDS plots. These plots are very different, they show the data differently. Please make sure you know which one you want (both can be applicable here, especially since you used Bray-Curtis dissimilarity) and change the wording. Since you already made the NMDS plots, I recommend fixing PCoA in the text.

See more here: https://www.davidzeleny.net/anadat-r/doku.php/en:pcoa_nmds

Paliy, O. and Shankar, V., 2016. Application of multivariate statistical techniques in microbial ecology. *Molecular ecology*, 25(5), pp.1032-1057.

Figure 3

The initial “a,b” is not needed because you distinguish the two later in the figure legend.

Figure 4

This is rather confusing because you highlight these four microbial phyla, but in the text state that you had connections to all the MAGs. “Most (98%) of the predicted host populations were assigned to 20 prokaryotic phyla each with more than 10 populations matched with their viruses.” Why were just these 4 highlighted? I see in the supplementary figure 1 there is the other networks. There is no explanation for this.

“b, Statistical distribution of modularity for the 20 virus-host networks compared with that of random matrices. Error bars denote 95% confidence intervals based on 1,000 randomisations. c, Linear regression relationships between the abundance of viruses and corresponding host phyla, indicated at the top of each plot.” Again you do not show the 20, only 4, the rest are in Supplementary Fig. 3. Why?

Figure 5

The letters for c and d are mixed up.

“Maximum-likelihood phylogenetic tree with phoH genes from AMD sediments (indicated by stars) compared to homologs found in eggNOG v5.0.0 database and the host proteins coloured by different phyla.” What do the outer colors denote? Different MAG phyla? Can you put the vOTU # in the star, so we can look into the virus-host linkage details more?

Reviewer #5:

Remarks to the Author:

This manuscript by Gao and co-workers (Patterns and ecological drivers of viral communities in acid mine drainage sediments) deals with the diversity and biogeography of viruses in AMD systems in China. The authors present a comprehensive analysis of the viral diversity and potential auxiliary metabolic genes. Not mentioned in the abstract is the additional work the authors put into analysing MAGs and linking viruses to potential hosts, also demonstrating a linear relationship between viral and host abundance for three major taxa. I particularly like how thoroughly most of the analysis were carried out and how the authors combine unsupervised machine learning (aka multivariate statistics) for unearthing the viral and host ecology. The distance decay relationship between viral genomic divergence and geographic distance is particularly satisfying in the manuscript.

Overall I only have a few major and a few minor comments that the authors should address prior to publication of this manuscript.

Major concerns:

a) The authors state in the title and in the last sentence of the abstract that this study represents a comprehensive analysis of viruses in AMD systems. However, the analyses are restricted to samples taken in China. I'd like to see an inclusion of public AMD data in here to provide evidence that these claims really hold true worldwide and are as generic as the authors claim.

b) The authors identified a lot of unknown viruses but the efforts to classify them properly are little. I'd like to see a better approach (e.g. identifying distantly related viral proteins in public databases and building phylogenetic trees) for these viruses to better understand the unknown viral diversity as this is the majority of the viruses present in the systems.

c) Although the authors made reads of the 16S rRNA gene amplicon data and the viral genomes available in SRA/NCBI, there is no information that neither the MAGs nor the reads of the metagenomes are publicly available. Without making the reads and the MAGs, i.e. all research data, publicly available, the study cannot be reproduced and should not be accepted for publication.

Minor comments:

- Line 59-60: This statement isn't true any longer. There have been plenty of studies out there that have used metagenomics to investigate the uncultivated viral diversity. This statement also contradicts the first sentence of your discussion.

- Line 63-67: There are other studies that looked at biogeography of viruses in ecosystems that the authors did not list, like the deep biosphere and soil. Please see for instance and include the references (and/or maybe others):

o Communications Biology volume 4, Article number: 307 (2021)

o Nature Communications volume 12, Article number: 4642 (2021)

o mSystems 2021 Volume 6 Issue 3 e00385-21

- Line 106-117: This part is really hard to read because it is (in contrast to the rest of the manuscript) built by first stating in one sentence the method, then in the second sentence the results. This paragraph could be more elegantly phrased so that the reader doesn't feel like they are reading material and methods.

- L142-149: This is another section in the manuscript that made me think that samples outside of China would be necessary to further explore the trend of the distance decay relationship.

- Figure 1, e: This panel needs a better explanation. Are "types" genes/proteins here? This panel is unclear to me.

Reviewer #6:

Remarks to the Author:

The authors use a partial structural equation modelling (pSEM) approach to analyse how different biotic and abiotic factors affect the viral community structure in acid mine drainage, aiming to separate the most likely structure of direct and indirect effects. Separate analyses are performed for the taxonomic and the functional composition of the viral community as the ultimate response variable in the pSEM model. In this, the community structure (taxonomic or functional) was

represented as the first principal component of a Principal Coordinate Analysis (PCoA).

Overall, this can be an appropriate approach for this type of complex data to unravel the structure of a complex set of predictor variables. However, I have several issues with the way the authors have applied these methods and interpreted the results in this case, that need improvement and clarification.

1) The authors are insufficiently clear on the technical details in the methods section on how the pSEM was actually developed. Classical SEM is based on the assumption that all relations in the model can be captured through a linear regression approach with normal error distributions. pSEM differs in that it allows for other types of linear models to be combined in an SEM framework, such as poisson regression or binomial regression. However, the authors do not mention at all what the different partial models are in their final model. It therefore remains unclear why a classical SEM (actually a path analysis, as latent variables were not used) would not work in this case. If a pSEM is used, the type of (hybrid) partial models should clearly be described.

2) I am concerned about the incredible high standardized path coefficients that the authors report in Fig. 3b. Their results show that temperature (MAT) is near-perfect predicting prokaryotic community composition, almost without remaining unexplained variation ($r=0.99$), while prokaryote composition is near-perfect in predicting viral community composition ($r=0.96$). If viral community composition can be directly derived from the prokaryote composition (so is hardly independent from that) is it then worthwhile to add such a complex analysis to the paper with 8 other predictor variables? The authors do not seem to discuss this. The same holds for the taxonomic viral composition (fig 3a) This seems to be directly derived from the prokaryote composition without much remaining variation ($r=1$). The authors should much better motivate why an SEM is appropriate in such a case.

3) The authors use a standard backward elimination approach to get to a set of predictors for viral composition. In addition however, an (p)SEM requires as input an expected causal structure of the predictor variables that is not strongly motivated here. In general when using (p)SEM, it is more interesting to test alternative hypothesis (eg based on conflicting theoretical predictions or empirical findings) rather than throwing everything in a single model and remaining 'everything significant'. In this case for example, it could have been evaluated if the prokaryote community mostly drives viral composition, or if viral composition is also subject to environmental drivers independent of drivers of prokaryote composition. So evaluate alternative SEM's and see which one is supported by the data best. In the current approach, there is always a result but that not necessary is the most interesting approach. Especially given the issue of the very high partial coefficients indicated above.

Han Olf, University of Groningen

Reviewer #1 (Remarks to the Author): In recent years it has been possible to begin to explore viral populations in different environments; in general, the microbiological study has been biased towards prokaryotes, leaving aside the effect that viral populations have on different communities and ecosystems. This work is very relevant because it allows having a broader panorama of the effects of the particular physicochemical conditions as well as the climatic ones from extensive sampling of the AMD of southern China, evaluating the interactions that viruses have with their hosts thus as the dynamics that arise in this type of ecosystems in particular.

As you well say in the discussion, it must be taken with caution to say how much the hosts influence the distribution of viruses, since although viruses cannot replicate in their absence, there is still a large number of viral sequences of which we do not yet know its role in the communities' dynamics and, on the other hand, host specificity has not been explored in depth.

Response: We thank you for your constructive comments. In the revised version, additional methods are used to explore the unresolved fraction of the viral diversity and functions, including the Lowest Common Ancestor (LCA) algorithm for classification of viral genomes and the VOG database for annotation of viral proteins (Fig. 1d and e, Line 106-117). Besides, we have conducted a more in-depth analysis of virus-host interactions, and highlighted the dynamics of the dominant phyla (*Proteobacteria* and *Thermoplasmata*) and their associated viruses in the acid mine drainage sediments (Fig. 4, Line 204-216). Despite these further efforts, caution must still be taken when interpreting the extent of influence that hosts have on viral distribution. This is stated in Line 339-343. All changes are marked in blue in the revised manuscript.

Reviewer #2 (Remarks to the Author): Gao et al. describe host virus linkages over a wide range of acid mine drainage sediments and focus on a case study of phosphorus acquisition auxiliary metabolic genes. This is a staggering dataset which by itself would be a great addition to the field. Moreover, the paper is very well written; it is clear, concise and cohesive. I really enjoyed reading it. All in all this work is valuable, promotes the field, the methodology is sound and i can imagine many more papers coming out of it. Most of my comments are minor and there are a few points in the discussion that i feel are overreaching, but otherwise my biggest concern that the metagenomic reads and MAGs are not publicly available.

Response: We thank you for your positive feedbacks. The metagenomic reads and MAGs are now available for download in the NCBI BioProject database with the accession no. PRJNA666025. Detailed accession numbers for individual reads and prokaryotic genomes are now listed in Supplementary Data 8 and Supplementary Data 9, respectively. You can also directly download the prokaryotic population genomes at <https://figshare.com/s/48f5d65bf4a4dfe4a476> (Line 573-580). The discussion is carefully refined/revised according to all of the Reviewers' comments.

Specific comments:

Line 78: Comics? I assume you meant omics

Response: This is corrected (Line 80).

Line 79: change strike to strive

Response: This is corrected as suggested (Line 82).

Line 143: in support of this

Response: This is corrected (Line 148).

Line 146: why did you choose a linear correlation? What i see is two unstructured clusters which lead to the same conclusion that there is a cutoff at 1 km for dissimilarity.

Response: Distance-decay relationship (DDR) typically uses a linear regression to describe community similarity decreases as the geographical distance increases (Wu et al., 2019, 4:1183–1195). Similarly, we choose a linear regression to show the distance-decay relationship of viral taxonomic and functional community dissimilarities at different spatial scales, i.e., local (pairwise distance ≤ 1 km), regional (pairwise distance > 1 km), and all scales (Line 148-154).

Figure 2e: in the legend the red line looks yellow

Response: This legend is now corrected (Line 862).

Lines 183-184: i would mention that populations were achieved by dereplicating MAGs.

Response: As reminded by Reviewer #3, and to keep consistency across all analyses in this manuscript, prokaryotic populations achieved by dereplicating metagenomic-assembled genomes (MAGs), instead of clustering 16S rRNA gene amplicon sequences, are now used to represent the prokaryotic communities. This is now clarified in Line 122-125.

Line 194: it's even more impressive to me that 13/20 have an $R^2 > 50$.

Response: As suggested by Reviewer #4, viral genomes < 10 kb are now discarded in our revised manuscript. However, re-analyses of the updated data sets resulted in virus-host abundance correlations generally similar to those described in our previous manuscript (Line 197-200).

Line 219: i would rephrase to say they were placed in a virus specific clade.

Response: Viral genes predicted from all identified viral genomes, instead of only representative viral genomes, were annotated with eggNOG database (v5.0.0) in the revised manuscript. As a result, 75 viral genes were assigned as *phoH*, including 4QCHF and COG0172. The predicted *phoH* genes, combined with their homologs from the recovered prokaryotic genomes and eggNOG v5.0.0 database, were used to construct a phylogenetic tree, and genes assigned as 4QCHF were mostly clustered with their counterparts from viruses and *Bacteroidota*, while genes assigned as COG0172 were mostly affiliated with homologs from *Proteobacteria* and *Patescibacteria* (Line 246-251 and Line 473-474).

Discussion: there are some seminal papers I don't see cited here, like Tyson and Banfield and Sun et al. also from the Banfield group.

Response: These references are now cited in Line 291 (References 38 and 39).

Line 249-251: our knowledge on any environmental system is incomplete. This is a moot point.

Response: This sentence is now rephrased (Line 288-291).

Line 251-257: the genes you are describing are the core genes that a virus cannot survive without. Of course they are universally distributed in viruses. Once again this is a moot point and should be replaced with a stronger conclusion.

Response: This part is now rewritten (Line 293-296).

Line 271: i doubt iron is limiting in these systems, but i was wondering if you were aware of the Ferrojan horse theory on vital attachment to iron receptors. It could be a cool link here.

Response: Thank you for reminding us about the Ferrojan horse theory. This paragraph is now rewritten accordingly (Line 312-325).

Line 275-281: does the pH in the sediment then release the viruses from the particles? Because otherwise being mineral attached would make them inactive.

Response: Increased pH could release viruses from the mineral particles with lower isoelectric point, and this point is now included in the discussion (Line 323-325).

Line 302-303: again this is a moot point. Of course viruses depend on their hosts for replication. What's interesting is discussing why the correlations mostly don't explain a huge part of the variability. It probably has to do with the broad taxonomic level of this analysis (e.g. Proteobacteria is a huge and incredibly diverse phylum), and probably with attachment to minerals.

Response: This sentence is revised in Line 346-350. Besides, we have conducted an additional in-depth analysis of virus-host interactions, and detailed virus-host dynamics of the dominant phyla, i.e., *Proteobacteria* and *Thermoplasmata*, and their associated viruses have now been highlighted and further discussed (Line 204-216 and Line 350-359).

Line 307-308: could you be more specific about these proposed phylum level infection mechanisms? That seems overreaching to me.

Response: Virus-host interaction structure across host phyla and sediment samples are now clarified in Fig. 5 (Line 226-235 and Line 359-363).

Line 309: you have no evidence of active viral replication. To do that you need either a time series or isotope studies. You have evidence of historical infections. Remove this sentence.

Response: This incorrect sentence is now deleted (Line 363).

Line 311: there are so many extreme environments not mentioned here, like the deep ocean, deserts and volcanic rock to name a few. I don't appreciate this attempt to spin the conclusions into a bigger frame, especially when there's no need of it. You have such a great dataset that spans AMD environments over space, pH etc. Also this sentence feels disconnected from the rest of the paragraph.

Response: This sentence is rephrased in Line 364-365.

Line 334: environments

Response: This is corrected in Line 394.

Line 434-447: this should be a separate section on MAG generation.

Response: As you suggested, this paragraph is now a separate section (Line 495-511).

Reviewer #3 (Remarks to the Author): The manuscript prepared by Gao et al. which was submitted to Nature Communications outlines an observational study wherein the authors examined the viral communities (and their hosts) extracted from acid mine drainage (AMD) sediments collected from mine sites around China. They explore host dynamics, virus dynamics and phosphorus functional genes to describe biogeography trends and community functions. It is not entirely clear to me why they used amplicons alongside the soil metagenome, though I presume it assisted in separating putative viruses from their hosts; not something we do in my own lab, but an interesting approach to the problem. The methods and analyses are sound throughout, and the narrative structure, writing, etc. is suitable for the subject matter. I have highlighted several places within the manuscript that would benefit from some revisions – sentences that are missing a word, or with poor sentence structure, or with odd word choices. I also recommend a slight expansion of the methods, particularly on the topic of sampling. Please see the file attached for grammatical corrections and further comments.

Response: We thank you for your constructive comments, which have been carefully addressed in the revised manuscript. To maintain consistency across analyses throughout the manuscript, in the revised version we recovered well-represented prokaryotic communities through genome-resolved metagenomics, and the results are generally comparable to those resolved by 16S rRNA gene amplicons. Besides, the Materials and methods section is now extended, particularly on the sampling procedures, the measurement of geochemical properties, and the identification of viral genomes. Other specific comments listed in the attached file have also been addressed carefully.

Line 28: Awkward phrasing.

Response: This sentence is revised (Line 29-30).

Line 40: Extreme AMD systems? Or AMD systems (which, compared to non-AMD systems, are extreme)?

Response: This is now revised (Line 41-42).

Line 80: Define this the first time it is used in the Introduction, not here.

Response: The word 'AMD' is defined in Line 79. Other abbreviations are also correctly defined throughout the manuscript.

Line 163: It's unclear which node this is referring to at first glance. Perhaps good to include 'ASVs' in the sentence.

Response: The prokaryotic richness is now clarified in Line 125-127.

Line 163-169: Unclear meaning; please split into simpler sentences.

Response: This sentence are rephrased (Line 169-178).

Line 170-172: Unclear meaning, please revise.

Response: We have conducted a major re-analysis of the data according to the Reviewers' comments (e.g., use of a 10 kb threshold for viral genomes, resolving prokaryotic communities through genome-resolved metagenomics, etc.). Thus, the whole paragraph is revised substantially (Line 169-182).

Line 242: Unclear meaning, please revise.

Response: This sentence is revised (Line 276-278).

Line 245-246: Awkward phrasing, please revise.

Response: This sentence is revised (Line 284-288).

Line 283-284: Awkward.

Response: This phrase is revised (Line 326-328).

Line 339: I'm also personally interested in the surrounding vegetation classification (e.g. tall open forest, etc.) and the 'normal' soil classification (e.g. chromic Solonetz [WRB], etc.), but that's probably a bit more trouble to find out.

Response: We did not record these information for the current study, but a typical AMD site is often associated with a large open area of mining waste or wasteland where the microbially mediated oxidative dissolution of sulfide minerals leads to the generation of acid drainage.

Line 340: How? Using a corer or a shovel or what? Was a single sample retrieved, or were many samples collected, bulked, and subsampled? How were sample points decided? That is, how did you prevent unconscious biases from affecting where you sampled (if you were aiming for random)? In contrast, what were the touchstone patterns to follow (if you were aiming for non-random)?

Response: For the 18 mine sites, ten sediment samples were collected at each site using a shovel from the top 10 cm of AMD sediments either at the centre or at ~1m from the edge of AMD ponds depending on safety and size of the ponds. However, only 90 high-quality DNA were obtained and used for further metagenomic sequencing. This has been clarified in Line 397-406 and Line 430-432.

Line 375-376: Awkward phrasing, please revise.

Response: This sentence is now revised (Line 434-435).

Line 394: Unclear meaning, please revise.

Response: This is now clarified (Line 447-448).

Line 397-401: Unclear meaning, please revise.

Response: This paragraph is now reorganized, revised and extended (Line 450-468).

Line 406-407: Unclear meaning, please revise.

Response: See our response above.

Line 410-412: Unclear meaning, please revise.

Response: This sentence is now revised (Line 467-468).

Line 462: ‘, respectively’ is only used for separate paired lists. That is, “A and B were classified as 1 and 2, respectively.” <- A is 1 and B is 2. If the pairing is adjacent, it isn’t required. That is, “A was classified as 1 and B was classified as 2.”

Response: ‘, respectively’ is deleted as suggested (Line 530).

Line 463: There’s a verb missing from this clause.

Response: This sentence is now revised (Line 528).

Line 472: Do ensure that you are following the correct procedure for citing software packages in Nature Communications. R has a proper citation (R Core Team), and the packages used within R also have proper citations.

Response: The R software and associated packages are now cited in the revised manuscript (Line 538-571).

Line 491-492: The word of interest here is ‘rate’ (single), not ‘DDRs’ (plural) – so the word to use is ‘was’ not ‘were’.

Response: This is corrected as suggested (Line 554).

Line 507-508: Awkward phrasing, please revise.

Response: This part is now revised substantially (Line 565-569).

Line 713-714: Considering the scale of the map, why bother including both? They overlap completely. Better to just include the sites, I think.

Response: Fig. 1a is now modified as suggested (Line 819-822).

Line 720-731: There is no need to make this an alphabet ordered list.

Response: The COG categories are listed by four functional types (Line 831-845).

Line 767-768: Unclear meaning, please revise.

Response: This sentence is now revised (Line 885-886).

Reviewer #4 (Remarks to the Author): In this paper, viral populations and their distributions were characterized in acid mine drainage sediments across China. These analyses came from 90 metagenomes in which MAGs were reconstructed and there was additional 16S rRNA gene amplicon data. Overall, it was enjoyable to read, you can tell the authors put in a large amount of effort, and it is very important to understand viral and microbial communities and their dynamics in anthropogenic environments. This was a lot of work and included great analyses determining ecological drivers of the viral communities based on abiotic and biological factors. For the most part, I was impressed to see extensive work was done to make these analyses top notch (e.g., ASVs over OTUs, metaG and Amplicon work, SILVA over Greengenes database, MAG reconstruction,...). This is also somewhat true for the virus work, except it was sad to see the 10kb threshold that is important for vOTUs not used (literature was cited which highlights the importance of the 10 kb threshold). This is worrisome but can be easily overcome if the analyses could be redone with the smaller contigs removed. But if the conclusions remain the same, then the current data could be kept with the extra work noted. The authors will have trouble making the case for vConTACT2 work, because the genera will look much different. This is a

minor part of the paper and could be removed if they do not want to do the re-analyses.

After reading the document a couple times, I realized that there are a lot of results and only some make their way into the discussion section. For example, figure 4. Part A shows some of the linkages (even though the figure says it shows all) and I am not sure what the reader is supposed to get from this. It would be great to know if there are more connections with specific phyla and how that may impact the environment. As it is now, we see dots connected. Part B Shows the abundances of vOTUs and MAGs in each sample. This is great but where is the interpretation? Part C shows what is in part B but now with all the samples combined. From looking at the data, I see some viruses correlate with hosts and some don't. It looks like regression analyses where thrown at each vOTU population connected to a phylum, no matter what it looked like. What are we supposed to get from this? Are some vOTUs suspected to have higher burst sizes, or do lytic infection more? It adds nothing to the paper as is. Part D is another way of showing the data in part b and c, with no interpretation. I could see parts b and d being useful if there was a targeted story to highlighting specific microbial phyla and how their dynamics with linked vOTUs may impact the environment. This is also done for figure 5. Here, I was pleased to see in the results a mention that there was "11 additional Burkholderiales populations" that were highlighted, but then no discussion.

I would caution the interpretation of the geography analyses in regard to distance from the equator because the samples come from only southern China and viruses are pulled from traditional metagenomes (as compared to a virome).

Some of the word choice is off and needs to be fixed, but overall the writing and flow are good. Viral populations were defined as vOTUs, but there is switching between writing it out and writing vOTU. After the first instance in which the abbreviation is given, please only use vOTU.

Response: We thank you for your thoughtful comments. These major concerns, including especially issues of the 10 kb threshold for vOTUs, presentation and discussion of virus-host interactions, and limitations of the geographic scale of sampling and utilized approaches (i.e., traditional metagenomics), have been carefully addressed in the revised manuscript. Specifically, a 10 kb threshold for viral genomes is used to re-analyse the data, and the resulted patterns are generally similar to those emerged using a 3 kb threshold. This is true even for the taxonomic composition (genera) generated using vConTACT2. Meanwhile, the dominant host phyla (*Proteobacteria* and *Thermoplasmata*) and their associated viruses are now highlighted to reveal virus-host abundance dynamics. Furthermore, the interpretation of the geography analyses and the potential roles of viral auxiliary metabolic genes are carefully revised to avoid overstating the results. In particular, the title of the manuscript and the discussion are modified to reflect the fact that sampling was limited in Southern China. All these changes are done largely through, but not limited to, the point-by-point responses to the specific comments below.

Lines 32-33 "The results demonstrated that host communities manipulate viral taxonomic and functional diversity"

I know what you mean by this and I agree but the word manipulate is poor word choice. What viruses are present, their diversity, and largely their abundance is dictated by host diversity and abundance. Please change manipulate to dictate, influence or a similar verb.

Response: The misleading word "manipulate" is changed to "dictate" as suggested

(Line 33).

Lines 36 “case analyses”

What does this mean? Do you mean a case as an example from your data? Please clarify and make the writing clearer.

Response: The word “case” is changed to “functional” (Line 37).

Lines 58-59 “and facilitating horizontal gene transfers (HGTs) via transduction.”

I would remove “via transduction” because they also transfer genes by lysing hosts, making those genes available for uptake (transformation).

Response: “via transduction” is removed as suggested (Line 60-61).

Lines 61-62 “While omics approaches have later been applied to explore viral diversity in the environment”

We are talking about metagenomics, metatranscriptomics and etc., so please say meta-omics or meta ‘omics and not omics. Omics is constantly used incorrectly because it alone means genomics, transcriptomics, and more, but not meta analyses.

Response: This word is corrected throughout the manuscript (Line 63, Line 80 and Line 289).

Line 64 “visual communities”

What do you mean by visual? Detected? Please clarify.

Response: This is a mistake, and it is now replaced with ‘viral communities’. (Line 68).

Line 78 “comics approaches”

I think you meant omics, but it should be meta-omics.

Response: This wrong word is corrected (Line 80).

Lines 79-81 “Here we strike to address this knowledge gap by utilizing a massive metagenomic data set generated from 90 acid mine drainage (AMD) sediments sampled across Southern China (Fig. 1a).”

Poor word choice with “strike”, please change.

Response: “strike” is now replaced with “strive” (Line 82).

Lines 83-88 “The biogeographic patterns of viruses in our data reveal strong impacts of prokaryotes, as well as both direct and indirect influence of abiotic factors on viral populations and functions. Moreover, extensive reconstruction of host genomes and prediction of virus-host interactions illustrated possible mechanisms for how viral communities are structured with respect to host population dynamics.”

The results and discussion should not be in the introduction, please move the writing or delete. Just lay out background on what has been done and what you did for context.

Response: These sentences are deleted and replaced with what we did to address the knowledge gap (Line 84-88).

Results

Line 91 “Overall viral diversity and novelty in the AMD sediments.”

Remove overall

Response: “Overall” is removed (Line 91).

Lines 97-99 “In total, we identified 7,442 potential viral populations (viral operational taxonomic units, vOTUs), which are suggested to approximately represent species-level taxonomy”

This is a lot and very exciting. Please make sure to list the contig cutoff, because in the previous sentence you give a bunch of ranges, so it is unclear. The cited paper gives guidelines of 10kb cut off for vOTUs.

Response: A 10 kb threshold is now applied to filter viral genomes, resulting in similar virus biogeographic patterns as we described in the previous draft. This threshold is now stated in Line 98 and Line 455.

Lines 113-115 “Functional composition of the viral communities was examined by comparing the predicted viral proteins against the eggNOG database (v5.0.0)24.”

This is good but in future work, I suggest using one or more of these: VOGDB (<https://vogdb.org/>),

MultiPhATE2 (<https://academic.oup.com/g3journal/article/11/5/jkab074/6178284?login=true>),

DRAM-v (<https://academic.oup.com/nar/article/48/16/8883/5884738>),

VIBRANT (<https://microbiomejournal.biomedcentral.com/articles/10.1186/s40168-020-00867-0>)

Response: the viral functions are now annotated with both the eggNOG v5.0.0 database and VOG v206 database (Line 112-117, Fig. 1e).

Lines 144-145 “Bray-Curtis similarities”

Please change similarities to dissimilarities. Bray-Curtis dissimilarity is used to quantify the differences in populations and is not the same as similarity. Dissimilarity = 1 - Similarity and can push populations together just because of how different they are to other populations.

Response: We used Bray-Curtis similarities here because the distance-decay relationships describe the pattern that similarity across communities decrease with increasing geographic distance across samples (Wu et al., 2019, 4:1183–1195) (Line 148-154).

Lines 178-182 “Extensive genome reconstruction was performed for the bacteria and archaea present in the sediments to link viral populations to their hosts. Finally, we recovered 7,759 prokaryotic MAGs (> 50% genome completeness and < 10% contamination), of which 3,948 were linked to 8,428 viral genomes.”

CPR biology - particularly host relationships - is very much an emerging field, with lots left to learn. Checkm on default parameters systematically underestimates CPR genome completeness. This is due to the fact that the program is not aware of the lineage specific losses of typical bacterial marker genes and thus the genomes appear to be less complete than they actually might be. The Checkm authors have provided a temporary work around using a smaller set of markers proposed by the Banfield lab: <https://github.com/Ecogenomics/CheckM/wiki/Workflows#using-cpr-marker-set>. In short, you might actually have some "high quality" CPR.

I did check the methods and read this “Prokaryotic population genomes were recovered from the 90 sediment samples using MetaBAT v2.12.162, MaxBin v2.2.263, Abawaca v1.0064, and Concoct v0.4.065 with default parameters, considering tetranucleotide frequencies, scaffolds coverage and GC content.”. This is extensive, which is great. Please do look into the CheckM comment and denote how many or high quality.

The next sentence starts with “finally” and this is odd because it is the third sentence and not part of a list.

Response: Thanks for your nice suggestions. The completeness and contamination of the recovered genomes assigned as CPR (*Patescibacteria*) are now assessed using a smaller set of markers proposed by the Banfield lab (Line 502-504), which increased the number of prokaryotic MAGs from 7,759 to 7,991 ($\geq 50\%$ genome completeness and $< 10\%$ contamination). Meanwhile, “Finally” is now deleted (Line 187).

Lines 184-186 “Most (98%) of the predicted host populations were assigned to 20 prokaryotic phyla each with more than 10 populations matched with their viruses.”

At lot of information is hidden in this sentence and it would be great to add in some more detail (unless there is a strict word limit). Most of the taxa are bacterial; please list # of bacteria phyla and # of archaeal taxa. I assume the latter part means there were 10 different vOTUs linked to each phylum? Expand on this. What percent of vOTUs and MAGs were linked? Was it with more than one method? Were many of the vOTUs linked to multiple MAGs?

Response: This part is now re-written to include these detailed information. Exact numbers (instead of percentages) of vOTUs and MAGs that are linked are given, together with their total numbers (Line 190-197). Meanwhile, “each with more than 10 populations matched with their viruses” means that there are at least 10 different prokaryotic populations in each of the 20 phyla being linked to viruses.

Lines 189-191 “We compared the networks with the null bipartite matrices with 95% confidence coefficient to evaluate the statistical significance of the modularity²⁵ and found that all networks displayed modular topology (Fig. 4b).”

For parts a and b in this figure only some of the data is shown and the rest is in different supplementary figure. This is not stated in the text or in the figure 4 legend. There is no explanation why a subset of the data was shown. I understand the lack of space, but you need to be transparent on why this data was selected and provide details of where to find the rest. Most readers do not look into the supplementary work unless there is something specific they are investigating. Please help the reader.

Response: As you suggested in the major comments, this part are now deleted and replaced with new results to show virus-host interaction structure across host lineages and sediment samples (Line 226-235, Fig. 5).

Discussion

Lines 240-243 “To bypass this hurdle, we adopted a total metagenome approach to uncover viral taxonomic and functional diversity in the AMD sediments, and generated saturated number of AMD viral genomes and genes (Fig. 1c).”

I think an “a” is missing between generated and saturated. It is also a little odd to say that a saturated number was generated. Technically we have no idea, but the abundance curves suggest these are well sampled for dsDNA viruses. We do know that metagenomes are bad at capturing viruses and I would add in a caveat in the text that a virome would likely capture more.

As an example, see:

Santos-Medellin, C., Zinke, L.A., Ter Horst, A.M., Gelardi, D.L., Parikh, S.J. and Emerson, J.B., 2021. Viromes outperform total metagenomes in revealing the spatiotemporal patterns of agricultural soil viral communities. *The ISME Journal*, pp.1-15.

Response: “a saturated number” is now rephrased (Line 280-281). The point that a

virome-based approach would likely capture more viral populations is added in Line 276-283.

Lines 273- “Previous investigations...”

This is great work, but you are missing other possibilities. What about viruses using iron for infection. Please incorporate some of this literature in the discussion:

Bonnain, C., Breitbart, M. and Buck, K.N., 2016. The Ferrojan horse hypothesis: iron-virus interactions in the ocean. *Frontiers in Marine Science*, 3, p.82.

Muratore, D. and Weitz, J.S., 2021. Infect while the iron is scarce: nutrient-explicit phage-bacteria games. *Theoretical Ecology*, pp.1-21.

Response: The Ferrojan horse hypothesis is now added in this section (Line 312-317).

Lines 308-310 “Additionally, based on lineage-specific VHRs, our results also revealed active viral replication and a possibly top-down control of prokaryotes via viral lysis in the AMD sediments (Fig. 4d).”

Metagenomes can show who is there and the potential, but alone it cannot show activity. Comparing viral and microbial advances from metagenomes does not describe activity, even if it was done over time or across different samples. Please remove this sentence.

If you wanted to determine this in future work, consider adding a metatranscriptome with a metaproteome, stable isotope probing, or BONCAT as a few examples.

Response: This sentence is deleted (Line 363). And thank you for suggesting the approaches for future activity studies.

Lines 321-323 “The identification of *phoH* genes in our AMD sediments provides evidence for the wide distribution of these viral AMGs in different habitats including extreme environments.”

The authors have done a great job of listing methodological limitations, but it is not done here. I appreciate that more P work beyond just *phoH* was done, but you need to bring dissenting literature in for context. The function of virally encoded *phoH* is not clear, and *phoH* expression in phosphate limited conditions appears to vary between hosts. The *phoH* gene is contentious because there is work showing that it may be involved in P cycling, but other work showing the gene is used in other ways and doesn't impact P cycling. There is no reference here highlighting the other work, which needs to be mentioned even if you spin to say this work will aid in disentangling or shed light on the truth. Below are some other works that I found on the *phoH* gene.

Sullivan et al. (2005): “Although the *phoH* gene is found widely distributed among both eubacteria and archaea [79], including all cyanobacteria, and is known to be induced under phosphate stress in *E. coli* [80], its function has not been experimentally determined. Bioinformatic analyses suggest that these *phoH* genes are part of a multi-gene family with divergent functions from phospholipid metabolism and RNA modification (COG1702 *phoH* genes) to fatty acid beta-oxidation (COG1875 *phoH* genes) [79].”

Zeng and Chisholm (2012) “*phoH* encodes an ATP binding protein with unknown function [15] and is considered a phosphate (*pho*) regulon gene because it is upregulated by P starvation in *E. coli* [16, 17]. Its expression is not upregulated during P starvation in marine cyanobacteria [2], however, suggesting that it may not play the same role as in *E. coli* [18]. Nonetheless, because of its prevalence in T4-like

cyanophages [9] and association with the pho regulon in *E. coli*, we examined its expression in our experiments. We found that the expression of *phoH* in the phage (Figures 1C and S1B) and host (Figure S1A) was not affected by P starvation, and therefore its role in both host and phage remains a mystery.”

Sullivan, M.B., Coleman, M.L., Weigele, P., Rohwer, F. and Chisholm, S.W., 2005. Three *Prochlorococcus* cyanophage genomes: signature features and ecological interpretations. *PLoS Biol*, 3(5), p.e144.

Zeng, Q. and Chisholm, S.W., 2012. Marine viruses exploit their host's two-component regulatory system in response to resource limitation. *Current Biology*, 22(2), pp.124-128.

Warwick-Dugdale, J., Buchholz, H.H., Allen, M.J. and Temperton, B., 2019. Host-hijacking and planktonic piracy: how phages command the microbial high seas. *Virology journal*, 16(1), pp.1-13.

Tetu, S.G., Brahamsha, B., Johnson, D.A., Tai, V., Phillippy, K., Palenik, B. and Paulsen, I.T., 2009. Microarray analysis of phosphate regulation in the marine cyanobacterium *Synechococcus* sp. WH8102. *The ISME journal*, 3(7), pp.835-849.

Lindell, D., Jaffe, J.D., Coleman, M.L., Futschik, M.E., Axmann, I.M., Rector, T., Kettler, G., Sullivan, M.B., Steen, R., Hess, W.R. and Church, G.M., 2007. Genome-wide expression dynamics of a marine virus and host reveal features of co-evolution. *Nature*, 449(7158), pp.83-86.

Response: We thank you for these useful comments and related articles. To support our assumption we have conducted an additional analysis of available P in the AMD sediments. The result is included in the revised manuscript (Line 251-255, Fig. 6b). Besides, we acknowledge the fact that functions other than P cycling have been documented for the *phoH* gene. This point is included in the discussion (Line 376-379).

Methods

Lines 341-343 “Samples were collected in 50 mL sterile tubes, kept in an icebox and transported to the laboratory, where they were stored at 4 °C prior to subsequent analyses.”

DNA degrades and microbes are active at 4 °C. The exact storage time needs to be given and if longer than 1-2 days, an explanation needs to be given. If several days, then this needs to be stated in the results section and some in the discussion.

Response: Sediments were stored at 4 °C and processed within 24 h. Each sediment was well mixed and divided into two fractions: one fraction for DNA extraction (subsequently stored at -80°C) and the other for physicochemical measurements (air-dried). The sample collection method is now revised (Line 397-406).

Lines 345-348 “Air-dried subsamples were analysed with standard methods for the determination of TOC (TOC-VCPH; Shimadzu, Columbia, MD), total nitrogen (TN) and total phosphorus (TP) (SmartChem; Westco Scientific Instruments Inc., Brookfield, CT).”

How many grams of sediment was used?

Response: More detailed information is added in the “Environmental measurements” section (Line 408-425).

Lines 358-359 “For the 90 samples,”

How much sediment was used?

Response: This is clarified (Line 427-429).

Lines 413-416 “The identified viral genomes were then clustered into vOTUs using the parameters 95% ANI and 85% alignment fraction of the smallest scaffolds, and the longest viral genome in each vOTU was chosen as the representative viral population genome.”

I am confused why all the correct thresholds from Roux et al. 2019 were used except the 10kb cutoff? I am seeing more of this in the literature without additional genome characterization to make sure contigs were not split or to determine true viral origin. More data is not always better and here it lowers the quality of the work. I would like to see the smaller vOTUs used, unless it can be shown that there are minimal differences.

Roux et al. 2015

“while providing near-perfect identification (>95% Recall and 100% Precision) on contigs of at least **10kb**.”

Roux et al. 2019

“Importantly, current methods for automatic virus sequence identification cannot reliably identify short (<10 kb) viral sequences, which should be interpreted with utmost caution.” This applies to the VirSorter version used in this paper.

The 10kb cutoff is critical for diversity analyses and vConTACT generation of genera. “*Some of these approaches require a minimum contig size—for example, contigs ≥ 10 kb for taxonomic classification based on gene content or diversity estimation—and will not be applicable to every genome fragment.”

Jang et al. 2019

vConTACT is not perfect but very good. It is largely limited to RefSeq viruses only. Applying it to vOTUs from metagenomes is questionable because it largely relies on the database size and contigs lengths. I would be okay with some conclusions if there was a 10kb cutoff because that is what vConTACT2 was evaluated on. “To evaluate scalability of our algorithm, we added 15,280 curated viral genomes and large genome fragments (≥ 10 kb) from the Global Ocean Virome (GOV) dataset to our reference network in 10% increments (that is, 0%, 10%, ..., 100% of the total dataset).”

Response: As suggested, we have re-analyzed our data using a 10 kb cutoff. While smaller numbers of viral genomes and vOTUs are identified (Line 97-99), the broad biogeographic patterns are similar to those emerged from a 3 kb threshold analysis. Meanwhile, similar viral genera are generated through application of vConTACT2 on the updated sets of genomes (≥ 10 kb) (Line 106-112).

Lines 475-478 “Bray–Curtis distances were used to construct the dissimilarity matrices for prokaryotic and viral community structure and function profiles, whereas Euclidean distances were calculated using environmental variables (vegan package v2.5-5).”

This is great but some minor rewording. Bray–Curtis dissimilarity can be used to identify distance, so something along this line is better “Bray–Curtis dissimilarity was used to construct to show distances for prokaryotic and viral community structure and function profiles, whereas Euclidean distances were calculated using environmental variables (vegan package v2.5-5).”

Response: This sentence is corrected as suggested (Line 541-544).

Line 490 “utilizing the Bray-Curtis distance”

Change distance to dissimilarity metric

Response: This is corrected (Line 553).

Line 498 “Bray-Curtis distance”

Change distance to dissimilarity metric

Response: This is corrected (Line 560).

Line 510 Data availability

What about the metagenomic reads and the MAG bins?

Response: NCBI accession numbers for individual reads and population genomes are now listed in Supplementary Data 8 and Supplementary Data 9, respectively. The prokaryotic population genomes can also be directly downloaded from figshare: <https://figshare.com/s/48f5d65bf4a4dfe4a476>. This is stated in Data availability (Line 573-580).

Figure 2

Throughout the paper part B and C are described as PCoA's but in fact they are NMDS plots. These plots are very different, they show the data differently. Please make sure you know which one you want (both can be applicable here, especially since you used Bray-Curtis dissimilarity) and change the wording. Since you already made the NMDS plots, I recommend fixing PCoA in the text.

See more here: https://www.davidzeleny.net/anadat-r/doku.php/en:pcoa_nmds

Paliy, O. and Shankar, V., 2016. Application of multivariate statistical techniques in microbial ecology. *Molecular ecology*, 25(5), pp.1032-1057.

Response: We replace the NMDS plots with PCoA plots (see Fig. 2b and c), and wording is now kept consistent throughout the manuscript (e.g. Line 857-859).

Figure 3

The initial “a,b” is not needed because you distinguish the two later in the figure legend.

Response: The initial “a,b” is now deleted as suggested (Line 867).

Figure 4

This is rather confusing because you highlight these four microbial phyla, but in the text state that you had connections to all the MAGs. “Most (98%) of the predicted host populations were assigned to 20 prokaryotic phyla each with more than 10 populations matched with their viruses.” Why were just these 4 highlighted? I see in the supplementary figure 1 there is the other networks. There is no explanation for this.

“b, Statistical distribution of modularity for the 20 virus-host networks compared with that of random matrices. Error bars denote 95% confidence intervals based on 1,000 randomisations. c, Linear regression relationships between the abundance of viruses and corresponding host phyla, indicated at the top of each plot.” Again you do not show the 20, only 4, the rest are in Supplementary Fig. 3. Why?

Response: This part is now replaced with new results (Fig. 4 and Fig. 5, Line 204-235). Specifically, the virus-host abundance correlations and interaction structure among the 20 prokaryotic phyla are now showed in Fig. 4a and Fig. 5a, respectively. Additionally, our results highlighted the dynamics of dominant phyla (*Proteobacteria* and *Thermoplasmata*) and their associated viruses.

Figure 5

The letters for c and d are mixed up.

“Maximum-likelihood phylogenetic tree with phoH genes from AMD sediments (indicated by stars) compared to homologs found in eggNOG v5.0.0 database and the host proteins coloured by different phyla.” What do the outer colors denote? Different MAG phyla? Can you put the vOTU # in the star, so we can look into the virus-host linkage details more?

Response: This figure is re-organized/revise, and the issues raised here are now corrected in the figure legend (Line 907-918).

Reviewer #5 (Remarks to the Author): This manuscript by Gao and co-workers (Patterns and ecological drivers of viral communities in acid mine drainage sediments) deals with the diversity and biogeography of viruses in AMD systems in China. The authors present a comprehensive analysis of the viral diversity and potential auxiliary metabolic genes. Not mentioned in the abstract is the additional work the authors put into analysing MAGs and linking viruses to potential hosts, also demonstrating a linear relationship between viral and host abundance for three major taxa. I particularly like how thoroughly most of the analysis were carried out and how the authors combine unsupervised machine learning (aka multivariate statistics) for unearthing the viral and host ecology. The distance decay relationship between viral genomic divergence and geographic distance is particularly satisfying in the manuscript.

Overall I only have a few major and a few minor comments that the authors should address prior to publication of this manuscript.

Major concerns:

a) The authors state in the title and in the last sentence of the abstract that this study represents a comprehensive analysis of viruses in AMD systems. However, the analyses are restricted to samples taken in China. I'd like to see an inclusion of public AMD data in here to provide evidence that these claims really hold true worldwide and are as generic as the authors claim.

Response: This is a very good suggestion. However, metagenomics analyses of AMD ecosystems have largely been focused on AMD solutions (e.g., ISME J. 2015, 9:1579-92; ISME J. 2015, 9:1280-94) and biofilms (especially those from the Richmond Mine in California, USA). Searching public databases only identified two studies that reported metagenomic sequencing of AMD sediments from other countries: Korzhenkov, et al., 2019, 7:11 with one sample from Parys Mountain (a copper mine in UK, SAMN09356021), and Mardanov et al., 2017, 5:e01355-17 with one sample from Kemerovo (a gold mine in Russia, SAMN07811784). Thus, we can not conduct such a meta-analysis but instead modify our manuscript including the title and discussion to avoid overstating (Line 2 and Line 299-301).

b) The authors identified a lot of unknown viruses but the efforts to classify them properly are little. I'd like to see a better approach (e.g. identifying distantly related viral proteins in public databases and building phylogenetic trees) for these viruses to better understand the unknown viral diversity as this is the majority of the viruses present in the systems.

Response: In the revised manuscript we use both vContact2 and the Lowest Common Ancestor (LCA) algorithm to classify the predicted viral genomes. Although

annotation rate is significantly improved, a large proportion of viral genomes is still unclassified (Fig. 1d, Line 106-112). This is largely attributable to the absence of complete genomes of viral isolates from AMD and associated environments. Similar results have been previously reported for meta-omics analyses of viral assemblies in other environments (e.g., Emerson et al., 2018, 3:870-880 and Daly et al., 2018, 4:352-361). We discuss this issue in the second paragraph of the Discussion (Line 284-296).

c) Although the authors made reads of the 16S rRNA gene amplicon data and the viral genomes available in SRA/NCBI, there is no information that neither the MAGs nor the reads of the metagenomes are publicly available. Without making the reads and the MAGs, i.e. all research data, publicly available, the study cannot be reproduced and should not be accepted for publication.

Response: All data from the current study are now deposited in NCBI and accession numbers for individual metagenome reads and population genomes are listed in Supplementary Data 8 and Supplementary Data 9, respectively (Line 573-580). The prokaryotic population genomes are also available through figshare: <https://figshare.com/s/48f5d65bf4a4dfe4a476>.

Minor comments:

- Line 59-60: This statement isn't true any longer. There have been plenty of studies out there that have used metagenomics to investigate the uncultivated viral diversity. This statement also contradicts the first sentence of your discussion.

Response: We now rephrase the sentences to reflect the historical development of viral ecology studies: traditionally dependent on cultivation-based methods but more recently using meta-omics approaches (Line 61-65).

- Line 63-67: There are other studies that looked at biogeography of viruses in ecosystems that the authors did not list, like the deep biosphere and soil. Please see for instance and include the references (and/or maybe others):

o Communications Biology volume 4, Article number: 307 (2021)

o Nature Communications volume 12, Article number: 4642 (2021)

o mSystems 2021 Volume 6 Issue 3 e00385-21

Response: These and other meta-omics studies, which explored viral diversity and dynamics especially in extreme environments, are now included in the Introduction (Line 77-80).

- Line 106-117: This part is really hard to read because it is (in contrast to the rest of the manuscript) built by first stating in one sentence the method, then in the second sentence the results. This paragraph could be more elegantly phrased so that the reader doesn't feel like they are reading material and methods.

Response: This part is now substantially revised to avoid repeating the method in the results (Line 106-117).

- L142-149: This is another section in the manuscript that made me think that samples outside of China would be necessary to further explore the trend of the distance decay relationship.

Response: A very limited number of AMD sediments metagenomes are available in

public databases. Please refer to our response to the ‘Major concerns (a)’.

- Figure 1, e: This panel needs a better explanation. Are “types” genes/proteins here? This panel is unclear to me.

Response: The COG categories of annotated viral proteins are grouped into four types, including information storage and processing (COG categories ABJKL), cellular processes and signaling (DMNOTUVWYZ), metabolism and transportation (CEFGHIPQ), and unknown functions. The figure legend of Figure 1e is now revised (Line 831-845).

Reviewer #6 (Remarks to the Author): The authors use a partial structural equation modelling (pSEM) approach to analyse how different biotic and abiotic factors affect the viral community structure in acid mine drainage, aiming to separate the most likely structure of direct and indirect effects. Separate analyses are performed for the taxonomic and the functional composition of the viral community as the ultimate response variable in the pSEM model. In this, the community structure (taxonomic or functional) was represented as the first principal component of a Principal Coordinate Analysis (PCoA).

Overall, this can be an appropriate approach for this type of complex data to unravel the structure of a complex set of predictor variables. However, I have several issues with the way the authors have applied these methods and interpreted the results in this case, that need improvement and clarification.

1) The authors are insufficiently clear on the technical details in the methods section on how the pSEM was actually developed. Classical SEM is based on the assumption that all relations in the model can be captured through a linear regression approach with normal error distributions. pSEM differs in that it allows for other types of linear models to be combined in an SEM framework, such as poisson regression or binomial regression. However, the authors do not mention at all what the different partial models are in their final model. It therefore remains unclear why a classical SEM (actually a path analysis, as latent variables were not used) would not work in this case. If a pSEM is used, the type of (hybrid) partial models should clearly be described.

2) I am concerned about the incredible high standardized path coefficients that the authors report in Fig. 3b. Their results show that temperature (MAT) is near-perfect predicting prokaryotic community composition, almost without remaining unexplained variation ($r=0.99$), while prokaryote composition is near-perfect in predicting viral community composition ($r=0.96$). If viral community composition can be directly derived from the prokaryote composition (so is hardly independent from that) is it than worthwhile to add such a complex analysis to the paper with 8 other predictor variables? The authors do not seem to discuss this. The same holds for the taxonomic viral composition (fig 3a) This seems to be directly derived from the prokaryote composition without much remaining variation ($r=1$). The authors should much better motivate why an sem is appropriate in such a case.

3) The authors use a standard backward elimination approach to get to a set of predictors for viral composition. In addition however, an (p)SEM requires as input an expected causal structure of the predictor variables that is not strongly motivated here. In general when using (p)SEM, it is more interesting to test alternative hypothesis (eg based on conflicting theoretical predictions or empirical findings) rather than throwing everything in a single model and remaining ‘everything significant’. In this case for example, it could have been evaluated if the prokaryote community mostly

drives viral composition, or if viral composition is also subject to environmental drivers independent of drivers of prokaryote composition. So evaluate alternative SEM's and see which one is supported by the data best. In the current approach, there is always a result but that not necessary is the most interesting approach. Especially given the issue of the very high partial coefficients indicated above.

Han Olf, University of Groningen

Response: We thank you for your thoughtful comments on our article, especially the piecewise structural equation modelling (pSEM) method. We applied pSEM in our previous analyses since a number of authors have proposed that the sample size should be considered in terms of the parameters to be estimated in classical SEM, a common ratio being 10:1 (e.g. Jackson, 2009, 10:128-141). With this criterion our sample size (90 sediments) is not large enough to perform SEM. Nevertheless, we have performed SEM analysis in our revised manuscript, and the SEM models provided satisfactory fit to our data, as suggested by the *P*-values (Chi-squared test) and root mean square error of approximation (RMSEA) (Fig. 3, Line 159-161).

Meanwhile, the SEM models revealed that mean annual temperature (MAT) had direct impacts on prokaryotic composition ($r = 0.72$, $P < 0.001$), which primarily drives viral taxonomic ($r = 0.94$, $P < 0.001$) and functional ($r = 0.96$, $P < 0.001$) composition (Fig. 3). The tight coupling between prokaryotic community and viral community was further corroborated by our host prediction analyses which demonstrated that almost all viruses exhibited parallel variations in abundance with their hosts (Fig. 4a and b, Line 184-216). At this stage the mechanisms underlying the highly strong influence of prokaryotes on viruses are not clear; but such patterns could at least be partly attributable to the parasitic lifestyle of viruses and methodological limitations of the current study (i.e. the viral genomes recovered from bulk metagenomes might be biased toward intracellular viruses). Noteworthy, other abiotic factors, especially distance from the equator, mean annual precipitation (MAP), and ferric iron, were also found directly impact viral populations and functions through the SEM analysis. All these aspects are now discussed in Line 297-325 and Line 339-343.

Besides, priori SEM models considering all theoretical or empirical mechanisms, instead of including all predictor variables, are now constructed (Supplementary Fig. 6, Line 560-564). Detailed R scripts used to construct the SEM models are now shared at <https://github.com/eco-gaoshao/viral-biogeography>, which is also declared in Code availability section in the revised manuscript (Line 582-584).

Reviewers' Comments:

Reviewer #1:

Remarks to the Author:

The response to the observations I make seems to me to be adequate. The manuscript is more robust, mainly due to the addition of methods for the diversity and function of viral genes. Specific comments: Correct the spelling of words such as analyze, organize, specialization, specialized, deionized.

Reviewer #2:

Remarks to the Author:

I am satisfied with the revised version of this manuscript. I still have a few very minor comments to the authors, but otherwise I think this paper can be accepted.

Figure 2 panel e missing the e label

Line 322 - delete the word "of"

Line 349 - Again, the fact that viruses need their hosts to replicate is not a paradigm, it's a fact. There's no need to emphasize a fact. Please throw this part of the sentence out.

Reviewer #3:

Remarks to the Author:

The authors have satisfactorily addressed my comments. Please see attached file for minor corrections.

Reviewer #4:

Remarks to the Author:

The authors have done an excellent job addressing my concerns. This work will be a great addition to the literature.

Reviewer #5:

Remarks to the Author:

The authors have answered all my queries and I congratulate them to this outstanding manuscript. Best, Alexander Probst

Reviewer #7:

Remarks to the Author:

In these comments I focus on the main issues raised by the previous reviewer who is no longer available to review the revision:

- (1) The authors do not mention at all what the different partial models are in their final model.
- (2) Concern about the high standardized coefficients reported and whether an SEM is appropriate given the high degree of intercorrelation among some of the predictors.
- (3) SEM requires as input an expected causal structure of the predictor variables that is not strongly motivated here. In general when using (p)SEM, it is more interesting to test alternative hypotheses.

The written responses by the authors seem to address these issues reasonably well. Part of the gap between reviewer expectations and author presentation seems to result from the fact that the authors are using SEM in exploratory mode (i.e., for explanatory modeling) while the reviewer is looking for something a bit more comparative (testing hypotheses). Both are valid enterprises and it seems the manuscript now provides more explanation for the logic behind the models examined and presented. I infer that the presentation is not very sophisticated in terms of anticipating all the methodological questions that reviewers may have, but may nonetheless be providing a useful contribution (given what these kinds of analyses are good for).

Reviewer #1 (Remarks to the Author):

The response to the observations I make seems to me to be adequate. The manuscript is more robust, mainly due to the addition of methods for the diversity and function of viral genes.

Specific comments: Correct the spelling of words such as analyze, organize, specialization, specialized, deionized.

Response: British English is now used throughout the manuscript.

Reviewer #2 (Remarks to the Author):

I am satisfied with the revised version of this manuscript. I still have a few very minor comments to the authors, but otherwise I think this paper can be accepted.

Figure 2 panel e missing the e label

Response: The label for panel e is now added.

Line 322 - delete the word "of"

Response: The word 'of' is now deleted (Line 328).

Line 349 - Again, the fact that viruses need their hosts to replicate is not a paradigm, it's a fact. There's no need to emphasize a fact. Please throw this part of the sentence out.

Response: The sentence is now deleted as suggested (Line 355).

Reviewer #3 (Remarks to the Author):

The authors have satisfactorily addressed my comments. Please see attached file for minor corrections.

Response: Reviewer #3's comments in the attached file were addressed in the previous round of review.

Reviewer #7 (Remarks to the Author):

In these comments I focus on the main issues raised by the previous reviewer who is no longer available to review the revision:

- (1) The authors do not mention at all what the different partial models are in their final model.
- (2) Concern about the high standardized coefficients reported and whether an SEM is appropriate given the high degree of intercorrelation among some of the predictors.
- (3) SEM requires as input an expected causal structure of the predictor variables that is not strongly motivated here. In general when using (p)SEM, it is more interesting to test alternative hypotheses.

The written responses by the authors seem to address these issues reasonably well. Part of the gap between reviewer expectations and author presentation seems to result from the fact that the authors are using SEM in exploratory mode (i.e., for explanatory modeling) while the reviewer is looking for something a bit more comparative (testing hypotheses). Both are valid enterprises and

it seems the manuscript now provides more explanation for the logic behind the models examined and presented. I infer that the presentation is not very sophisticated in terms of anticipating all the methodological questions that reviewers may have, but may nonetheless be providing a useful contribution (given what these kinds of analyses are good for).

Response: Thank you for reminding us about the deficiency in our response to Reviewer #6's comments. To be more specific, we respond to the three main issues as follows:

(1) The classical SEM uses a linear regression model with normal error distributions was applied in our revised manuscript. Thus, our final models do not incorporate different partial models.

(2) The SEM models confirmed the highly strong and undisputed influence of prokaryotes on viruses as revealed in Fig. 2a. We performed the SEM analysis in order to simultaneously test the causal links among variables, and to quantify both direct and indirect effects of biotic and abiotic factors on viral communities. This is clarified in Line 157-160.

(3) To use SEM in comparative mode, we constructed and tested the priori models as Reviewer #6 suggested (Supplementary Fig. 2). Compared with the priori models, the final SEM models could better support the data, and the description of discrepancies between the priori models and the final SEM models are now added in Line 160-167.